# *Perilla frutescens*: A Rich Source of Pharmacological Active Compounds

**DOI:** 10.3390/molecules27113578

**Published:** 2022-06-02

**Authors:** Tianyu Hou, Vasudeva Reddy Netala, Hongjiao Zhang, Yun Xing, Huizhen Li, Zhijun Zhang

**Affiliations:** 1School of Chemical Engineering and Technology, North University of China, Taiyuan 030051, China; nvasu7kt@gmail.com (V.R.N.); b1604012@st.nuc.edu.cn (H.Z.); hzli@nuc.edu.cn (H.L.); 2Jinzhong Institute of Industrial Technology and Innovation, North University of China, Jinzhong 030600, China; 3Graduate School of Humanities, Nagoya University, Nagoya 4648601, Japan; sxcjxy8807@yahoo.co.jp

**Keywords:** *Perilla frutescens*, secondary metabolites, bioactive compounds, pharmacological activities

## Abstract

*Perilla frutescens* (L.) Britton, an important pharmaceutical and nutraceutical crop, is widely cultivated in East Asian countries. In this review, we present the latest research findings on the phytochemistry and pharmacological activities of *P. frutescens*. Different databases, including PubMed, Scopus, CNKI, Agricola, Scifinder, Embase, ScienceDirect, DOAJ, and Web of Science, were searched to present the best review. In this review, we clearly represent the active constituents responsible for each and every pharmacological activity, plausible mechanism of action, and maximum inhibitory concentrations, as well as IC_50_ values. Approximately 400 different bioactive compounds, including alkaloids, terpenoids, quinines, phenylpropanoids, polyphenolic compounds, flavonoids, coumarins, anthocyanins, carotenoids, neolignans, fatty acids, polycosanols, tocopherols, and sitosterols, have been reported in the leaves, seeds, roots, and aerial parts of *P. frutescens*. The bioactive constituents of *P. frutescens* exhibited different enzyme-inhibition properties, including antihyaluronidase effects and aldose reductase inhibitory, α-glucosidase inhibitory, xanthine oxidase inhibitory, and tyrosinase inhibitory properties. *P. frutescens* showed strong anti-inflammatory, antidepressant, anti-spasmodic, anticancer, antioxidant, antimicrobial, insecticidal, neuroprotective, and hepatoprotective effects. Hence, the active constituents of *P. frutescens* used in the treatment of diabetes and diabetic complications (retinopathy, neuropathy, and nephropathy), prevention of hyperuricemia in gout patients, hyper pigmentation, allergic conditions, skin inflammation, skin allergy, atopic dermatitis, periodontosis, androgenic alopecia, gastric inflammation, oesophagitis, carcinogenesis, cardiovascular, Alzheimer’s, Parkinson’s, and cerebral ischemic disorders. Furthermore, we revealed the most active constituents and possible mechanisms of the pharmacological properties of *P. frutescens*.

## 1. Introduction

*Perilla frutescens* (L.) Britton, belonging to the family Lamiaceae, is widely distributed in East Asian countries, such as Japan, China, Korea, and Vietnam. Since ancient times, it has been used in traditional Chinese medicine and is cultivated as an edible crop in mainland China, Japan, and Korea. The leaves of Perilla were used in preparing vegetable curries, pickles, and chutneys. Several studies have reported the presence of rich polyphenolic compounds, which exhibit high antioxidant capacity, in Perilla leaves. Green tea prepared with Perilla leaves is highly popular in East Asian countries, due to its high antioxidant capacity, and has been used to treat fish and crab poisoning in these countries. Perilla essential oil (PEO), extracted from *P. frutescens* leaves, is a complex mixture of volatile components, constituting approximately 150 to 200 different compounds, exhibiting high antioxidant, anticancer, anti-inflammatory, insecticidal, and antimicrobial activities; hence, it is mainly used in processed meat, bakery products, frozen foods, puddings, soups, and preservatives. Perilla oil, extracted from seeds, was regularly used as an edible oil in Mainland China. Modern medical research has revealed that different plant parts of *P. frutescens* possess enormous amounts of bioactive secondary metabolites, including terpenoids, flavonoids, alkaloids, steroids, quinines, and phenolic compounds, which exhibit a wide range of biological activities and have immense potential applications as pharmaceuticals, nutraceuticals, agrochemicals, biopesticides, flavours, fragrances, colours, and food additives. Different biological activities, including anti-allergic, anti-depressant, hypolipidemic, hepatoprotective, neuroprotective, anti-inflammatory, anticancer, antioxidant, and antimicrobial activities, reported in *P. frutescens* were attributed to the presence of bioactive secondary metabolites in the different plant parts. In this review, we aim to present up-to-date research on *P. frutescens*. The present review reveals practically all compounds/metabolites present in different plant parts of *P. frutescens*, and discusses different biological activities of *P. frutescens*, mechanisms of biological activities, and compounds involved in the biological activities of *P. frutescens*.

## 2. Phytoconstituents of *P. frutescens*

Different types of phytoconstituents, including alkaloids, phenylpropanoids, terpenoids (monoterpenes, diterpenes and triterpenoids), phenolic acids, flavonoids (flavones, flavonols, flavanones, isoflavanones, aurones and chalcones), anthocyanins, coumarins, carotenoids, neolignans, fatty acids, policosanols, tocopherols, sitosterols, glycosides, glucosides, peptides and benzoxipen derivatives, have been reported from the seeds, leaves, and aerial parts of *P. frutescens* (Table 1).

### 2.1. Alkaloids, Phenylpropanoids, and Terpenoids

Wang et al. [8] reported that the aerial parts of *P. frutescens* contain an important alkaloid, neoechinulin A, which inhibits mitogen-activated protein kinase (MAPK) phosphorylation. Other alkaloids present in the aerial parts include 1H-indole-3-carboxylic acid and indole-3-carboxaldehyde. Phenylpropanoids, organic compounds mainly present in plants, are biosynthesised from phenylalanine or tyrosine, via the shikimate pathway. Many phenylpropanoids, also, act as precursors or intermediate materials for the biosynthesis of flavonoids, coumarins, and lignins. The leaves and other aerial parts of *P. frutescens* contain various important phenylpropanoids, such as elemicin, isoelemicin, myristicin, and methylisoeugenol. Elemicin and myristicin are widely accepted psychoactive compounds. Isoelemicin and methylisoeugenol exhibit antiplasmodial effects against *Plasmodium falciparum* strains. The leaves, also, contain antidepressant and antioxidant phenylpropanoids, including dillapiole, nothoapiole, perilloside E (phenylpropanoid glucoside), and allyltetramethoxybenzene. Phenylpropanoids, such as elemicin, isoelemicin, dillapiole, nothoapiole, and allyltetramethoxybenzene, have also been found to effectively inhibit proinflammatory cytokines during lung inflammation. Various terpenoids in the leaves and other aerial parts of *P. frutescens* include monoterpenoids, sesquiterpenoids, and triterpenoids. Monoterpenoids of *P. frutescens* are of the perilla-ketone and perillaldehyde type. Perilla-ketone-type monoterpenoids include frutescenone A, frutescenone B, frutescenone C, isoegomaketone, and 9-hydroxyisoegomaketone [54,55] which exhibit anti-inflammatory activity, by inhibiting the expression of proinflammatory cytokines. Perillaldehyde-type monoterpenoids include 3-hydroxyperillaldehyde and perillic acid [56]. Furthermore, leaves and other aerial parts of *P. frutescens*, also, contain acyclic, monocyclic, and bicyclic monoterpenoids. Acyclic monoterpenoids include β-myrcene, geraniol, β-citronellene, nerol, linalool, and ocimene. Monocyclic monoterpenoids include terpineol, thymol, phellandrene, 1,8-Cineole, damascenone, terpinene, carvacrol, carvone, piperitone, piperitenone, limonene, menthone, carveole, and pulegone [1,5,7]. Bicyclic monoterpenoids present in the leaves include δ -2-carene, camphene, verbenol, and sabinene. *P. frutescens* leaves are rich in sesquiterpenoids, which include acyclic, monocyclic, bicyclic, and tricyclic types. Acyclic sesquiterpenoids include farnesene, farnesol, and nerolidol. Monocyclic sesquiterpenoids include α-humulene, bisabolene, germacrene, elemene, and β-ionone [22]. The leaves are, particularly, rich in bicyclic and tricyclic sesquiterpenoids. Bicyclic sesquiterpenoids include α-pinene, β-pinene, β-caryophyllene, ε-muurolene, α-cadinene, β-cadinene, α-santalol, α-bulnesene, β-gurjunene, β-selinene, α-fenchene, α-cadinol, eremophilene, calarene, and valencene. Tricyclic sesquiterpenoids include spathulenol, viridiflorene, cubebene, alloaromadendrene, patchoulane, α-copaene, longifolen, and ylangene. To date, only one diterpenoid phytol has been reported from the leaves and aerial parts of *P. frutescens*. There are about 200 different types of volatile components that have been reported from the leaves and other aerial parts of *P. frutescens* (Table 2). The leaves of *P. frutescens*, also, constitute pentacyclic triterpenoids, including ursolic acid, corosolic acid, 3-epicorosolic acid, pomolic acid, tormentic acid, hyptadienic acid, oleanolic acid, augustic acid, and 3-epimaslinic acid [14], which exhibit cytotoxicity against various cancers, including leukaemia, breast, and hepatic carcinomas.

### 2.2. Polyphenolic Compounds

The leaves, stems, and seeds of *P. frutescens* are rich in different types of phenolic compounds. The leaves constitute rosmarinic acid, methylrosmarinic acid, caffeic acid and its derivatives including ethyl caffeate, methylcaffeate, vinyl caffeate, trans-p-menthenyl caffeate, caffeic acid-3-*O*-glucoside, (Z, E)-2-(3,4-dihydroxyphenyl)ethenyl ester of caffeic acid, and (Z,E)-2-(3,5-dihydroxyphenyl)ethenyl ester of caffeic acid [25]. The leaves, also, contain protocatechuic acid, protocatechuic aldehyde, chlorogenic acid, isovanillic acid, sinapic acid, gallic acid, ferulic acid, 4-coumaric acid, coumaroyl tartaric acid, 4-hydroxyphenyl lactic acid, sagerinic acid, p-hydroxybenzoic acid, and hydroxytyrosol. The phenolic compounds, present in *P. frutescens* seeds, include rosmarinic acid, methyl rosmarinic acid, rosmarinic acid-3-*O*-glucoside, 3′-dihydroxyl-rosmarinicacid-3-*O*-glucoside, caffeic acid, caffeic acid-3-*O*-glucoside, vanillic acid, ferulic acid, and cimidahurinine [23]. *P. frutescens* stems contains caffeic acid and their derivatives, mainly ethyl caffeate, methyl caffeate, vinyl caffeate, and protocatechuic aldehyde.

### 2.3. Flavonoids

The leaves, stems, fruits, and seeds of *P. frutescens* constitute different types of flavonoids, including flavones, flavanones, chalcones, and aurones. *P. frutescens* leaves contain flavones, such as luteolin, apigenin, scutellarein, negletein, vicenin-2, and catechin [17,28,40] along with several derivatives of luteolin, apigenin, and scutellarein [38] which include luteolin-7-*O*-glucuronide, luteolin-7-*O*-diglucuronide, luteolin 7-0-glucuronide -6″-methyl ester, luteolin-7-*O*-glucoside, apigenin-7-*O*-glucuronide, apigenin-7-*O*-diglucuronide, apigenin-7-*O*-glucoside, apigenin 7-*O*-caffeoylglucoside, scutellarein-7-*O*-glucuronide, and scutellarein 7-*O*-diglucuronide. The flavones in *P. frutescens* fruits include luteolin, apigenin, and chrysoeriol, while those in seeds include luteolin, luteolin-5-*O*-glucoside, luteolin-7-*O*-glucoside, apigenin, apigenin-7-*O*-glucoside, diosmetin, chrysoeriol, and catechin [22]. The flavanones in *P. frutescens* leaves include shisoflavanone A, liquiritigenin, 5,8-dihydroxy-7-methoxyflavanone, (2S)-5,7-dimethoxy-8,4′-dihydroxy flavanone, and 8-hydroxy-5,7-dimethoxyflavanone. The chalcones in *P. frutescens* leaves include 2′,4′-dimethoxy-4,5′,6′-trihydroxychalcone, and 2′,3′-dihydroxy-4′,6′-dimethoxychalcone, along with an aurone, (Z)-4,6-dimethoxy-7,4′-dihydroxyaurone [41].

### 2.4. Anthocyanins, Coumarins, Carotenoids, and Neolignans

*P. frutescens* leaves contain anthocyanins and their glucosides, which include shisonin, cis-shosinin, malonyl shisonin, cis-malonyl shisonin, cyanidin 3-*O*-feruloyl glucoside-5-*O*-glucoside, cyanidin 3-*O*-caffeoyl glucoside-5-*O*-glucoside, cyanidin 3-*O*-caffeoyl glucoside-5-*O*-malonyl glucoside, and peonidin 3-*O*-malonyl glucoside-5-*O*-p-coumarylglucoside [44]. *P. frutescens* leaves constitute esculetin and 6,7-dihydroxycoumarin, which possess xanthine oxidase (XO)-inhibitory and anti-inflammatory activities. The leaves constitute carotenoids, such as loliolide and isololiolide, which, also, possess XO-inhibitory activity [20] Two neolignans in *P. frutescens* leaves, magnosalin, and amanicin, have the potential to treat endotoxemia and inflammation [46].

### 2.5. Fatty Acids, Policosanols, Tocopherols, and Sitosterols

Perilla seed oil constitutes 40% of the total seed weight. *P. frutescens* seed oil contains saturated fatty acids, such as lauric acid, myristic acid, pentadecanoic acid, palmitic acid, heptadecanoic acid, stearic acid, arachidic acid, and behenic acid. The unsaturated fatty acids in the seeds include monounsaturated fatty acids (MUFAs), such as palmitoleic acid (16:1 cis-7), oleic acid (18:1 cis-9), eicosenoic acid (20:1 cis-11), and polyunsaturated fatty acids (PUFAs) such as linoleic acid (18:2 cis-9,12), α-linolenic acid (18:3 cis-9,12,15), eicosadienoic acid (20:2 cis-11,14), and eicosatrienoic acid (20:3 cis-11,14,17). The perilla seed oil is, mainly, rich in unsaturated fatty acids and is constituted by 54–64% of α-linolenic acid, 14–23% of oleic acid, 11–16% of linoleic acid, and 7–8% of saturated fatty acids. Furthermore, *P. frutescens* seed oil contains long-chain alcohols, known as policosanols. *P. frutescens* seeds are highly rich in policosanols and contain 427.83 milligrammes of policosanols per kilogramme of perilla seed oil. Policosanols present in *P. frutescens* seed include eicosanol, heneicosanol, docosanol, tricosanol, tetracosanol, pentacosanol, hexacosanol, heptacosanol, octacosanol, nonacosanol, and triacontanol. Tetracosanol, hexacosanol, and octacosanol account for 88% of all policosanols. *P. frutescens* seeds, also, contain important vitamin-E-related antioxidant compounds, known as tocopherols, which include δ-tocopherol, γ-tocopherol, β-tocopherol, and α-tocopherols. The γ-tocopherol content in perilla seeds was, approximately, equal to that of α-linolenic acid and constitutes 70–80% of the total tocopherol content. The seeds, also, contain phytosterols, such as campesterol, stigmasterol, β-sitosterol, β-amyrin, β-cholestanol, and 5α-cholestane.

### 2.6. Glucosides, Peptides, Benzoxepin Derivatives, and Other Constituents

*P. frutescens* leaves, reportedly, contain different types of glucosides, including dehydrovomifoliol, perillanolide A, perillanolide B, perilloside A, perilloside B, perilloside C, perilloside D (monoterpene glucosides) [20] loganin (iridoid glucoside), 5′-β-d-glucopyranosyl oxyjasrnonic acid, 3-β-d-glucopyranosyl-3-epi-2-isocucurbic acid, 3-β-d-glucopyranosyloxy-5-phenylvalericacid, n-octanoyl-β-d-fructofuranosyl-α-d-glu- copyranoside, and 4-(3,4-Dihydroxybenzoyloxymethyl)phenyl-*O*-β-d-glucopyranoside (polyphenolic glucoside). The leaves, also, contain five β-d-glucosides (eugenyl-β-d-glucoside, benzyl-β-d-glucoside, β-sitosteryl-β-d-glucoside, prunasin, and sambunigrin) and methyl-α-d-galactoside [19]. The seeds contain oligopeptides (PSO), and dipeptides, such as Tyr-Leu and Phe-Tyr. The glycoprotein Pf-gp6 has, also, been reported in *P. frutescens* leaves. *P. frutescens* stems contain benzoxepin derivatives, such as perilloxin and dehydroperilloxin. Other constituents, such as p-hydroxybenzaldehyde, p-hydroxyacetophenone, trans-p-hydroxycinnamic acid, and 3,4,5-trimethoxycinnamyl alcohol, are reportedly present in *P. frutescens* leaves [20].

## 3. Biological Functions of *P. frutescens*

### 3.1. Aldose Reductase Inhibitory Activity

Aldose reductase, or aldehyde reductase (AR), an important NADPH-dependent enzyme, is, ubiquitously, present at higher concentrations in the heart, eyes, neurons, and kidneys. This enzyme catalyses the reduction in monosaccharides, into their respective sugar alcohols. AR converts glucose into sorbitol, via the polyol pathway of glucose metabolism [61]. Excessive sorbitol is severely damaging, and its accumulation in the eyes, neurons, and kidneys of diabetic patients causes retinopathy, neuropathy, and nephropathy, respectively [62]. AR inhibitors are a class of drugs that averts sorbitol accumulation, by preventing or delaying the action of AR in tissues such as eyes, kidneys, and neurons, and are, consecutively, used to prevent the corresponding diabetic complications. Different bioactive compounds from plants possess AR inhibitory activity [63]. The rosmarinic acid, chlorogenic acid, caffeic acid, protocatechuic acid, and methyl rosmarinic acid contained in *P. frutescens* leaves possess significant AR-inhibitory activity. Figure 1 represents the different phytoconstituents of *P. frutescens* involved in aldosereductase-inhibitory activity. Rosmarinic acid exhibited strong inhibition against AR, with an IC_50_ of 2.77 µM, followed by chlorogenic acid (IC_50_ = 3.16 µM) and methyl rosmarinic acid (IC_50_ = 4.03 µM). However, caffeic acid and protocatechuic acid exhibited weak AR-inhibitory activity (Paek et al., 2013). Luteolin, apigenin, and diosmetin, isolated from *P. frutescens* seeds, also, showed AR inhibitory activity. Luteolin was found to be a strong AR inhibitor, with an IC_50_ of 1.89 µM, followed by apigenin (IC_50_ = 4.18 µM) and diosmetin (Lee et al., 2016). Different glycosides isolated from *P. frutescens* leaves, including four monoterpene glycosides (perilloside A-D), five β-d-glucosides (eugenyl-β-d-glucoside, benzyl-β-d-glucoside, β-sitosteryl β-d-glucoside, prunasin, and sambunigrin), and methyl-α-d-galactoside, significantly inhibited AR. In particular, perilloside A and perilloside C have been proven to be potent AR inhibitors that show competitive inhibition, as demonstrated by L–B plots [19]. Caffeic acid-3-*O*-glucoside and rosmarinic acid-3-*O*-glucoside isolated from *P. frutescens* seeds also exhibited AR inhibitory activity. Thus, these bioactive constituents in *P. frutescens* may be useful as a remedy to treat diabetic complications.

### 3.2. α-Glucosidase Inhibitory Activity

α-glucosidase is a hydrolytic enzyme that breaks down starch, oligosaccharides, and disaccharides into simple sugars, such as glucose molecules, by acting on α (1–4) bonds to facilitate intestinal absorption of carbohydrates. Their action drastically increases post-meal blood sugar levels in diabetes patients. α-Glucosidase inhibitors prevent or delay the breakdown of carbohydrates into simple sugars (glucose molecules), which delays the intestinal absorption of glucose, and consequently the increase in blood sugar levels after meals [64]. Hence, α-glucosidase inhibitors have been successfully used in treating type II diabetes mellitus. Figure 1 show the bioactive compounds involved in α-glucosidase inhibitory activity of *P. frutescens*. Ha et al. [23] reported that the five phenolic compounds isolated from *P. frutescens* seeds exhibited α-glucosidase inhibitory activity in a dose-dependent manner. Luteolin was found to be a potent α-glucosidase inhibitor with an IC_50_ value of 45.4 µM, and inhibited α-glucosidase in a non-competitive manner with an inhibition constant (K_I_) of 45.0 µM and Michaelis-Menton’s constant (Km) of 259.3 µM. Caffeic acid-3-*O*-glucoside, rosmarinic acid, rosmarinic acid-3-*O*-glucoside, and apigenin exhibited α-glucosidase inhibitory activity at concentrations higher than 100 µM. Owing to their α-glucosidase inhibitory activity, bioactive compounds of *P. frutescens* can be formulated into oral antidiabetic medications.

### 3.3. Xanthine Oxidase (XO) Inhibitory Activity

Xanthine oxidase or xanthine oxidoreductase (XO) enzyme plays an important role in purine catabolism, wherein, it catalyses the oxidative hydroxylation step; hypoxanthine is first converted to xanthine which is further converted to uric acid. Hence, the XOinhibitors significantly reduce uric acid production when treating gout and hyperuricemia [65] Oxidative hydroxylation step catalysed by XO produces reactive oxygen species (ROS) such as superoxide (O_2_^−^.) and hydrogen peroxide (H_2_O_2_) radicals. ROS cause oxidative stress, which is associated with several diseases such as cardiovascular, neurological, and aging. Hence, XO inhibitors play a role in ROS inhibition [66,67]. Synthetic XO inhibitors cause adverse effects such as anaphylactic shock, Stevens-Johnsonsyndrome, hepatotoxicity, and epidermal necrolytic effects [68,69]; hence, this necessitates isolating natural XO inhibitors from different plant sources. Different bioactive compounds isolated from *P. frutescens* leaves showed potent XO-inhibitory activity (Figure 1). Two caffeic acid esters, (Z-E)-2-(3,4-dihydroxyphenyl) ethenyl ester and (Z-E)-2-(3,5-dihydroxyphenyl) ethenyl esters exhibit potent XO inhibitory activity with IC_50_ values of 0.021 and 0.121 µg/mL, respectively. In particular, (Z-E)-2-(3,4-dihydroxyphenyl) ethenyl ester exhibited XO inhibition equal to that of the standard drug, allopurinol. The L–B plots demonstrated that these caffeic acids exhibited non-competitive inhibition [70]. A chalcone,2′,4′-dimethoxy-4,5′,6′-trihydroxychalcone and a flavone, luteolin (3′,4′,5,7-Tetrahydroxyflavone) possess XO inhibitory activity with IC_50_ values of 0.21 and 2.18 µM, respectively. The L–B plots demonstrated that chalcone exhibited mixed-type inhibition, whereas luteolin exhibited competitive inhibition [41]. Furthermore, flavonone, ((2S)-5,7-dimethoxy-8,4′-dihydroxyflavanone); coumarin, esculetin (6,7-dihydroxycoumarin); and scutellarein, (4′,5,6,7-Tetrahydroxyflavone) also exhibited good XO inhibitory activities with IC50 values of 18.44, 32.56, and 48.66 µM, respectively. The Lineweaver–Burk plots indicated that flavonone and scutellarin exhibited mixed-type inhibition, while esculetin displayed competitive inhibition. An aurone, ((Z)-4,6-dimethoxy-7,4′-dihydroxyaurone), also, presented good XO inhibition. Negletein (5,6-dihydroxy-7-methoxyflavone), scutellarein-7-glucuronide (breviscapin), sericoside (triterpene), loliolide, isololiolide (carotenoids), dehydrovomifoliol, perillanolide A, perillanolide B (monoterpeneglycosides), 4-(3,4-dihydroxybenzoyloxymethyl) phenyl-*O*-β-d-glucopyranoside, trans-p-hydroxy cinnamic acid, p-hydroxybenzaldehyde, and p-hydroxyacetophenone, also, exerted XO-inhibitory activities, with IC_50_ values greater than 200 µM [17]. Thus, the bioactive compounds of *P. frutescens* could be used in treating hyperuricemia in patients with gout.

### 3.4. Tyrosinase Inhibitory Activity

Tyrosinases are oxidases involved in the initial stages of melanin biosynthesis. Overproduction of melanin causes hyper pigmentation and other skin disorders, such as melasma, solar melanosis, senile lentigos, and ephelides [71]. The modern lifestyle, junk foods, insomnia, industrial smoke, pollution, and synthetic drugs cause oxidative stress, which leads to aging and skin disorders. Hence, development of natural tyrosinase inhibitors with antioxidant capacity, to reduce hyper pigmentation, aging, and skin disorders, is essential. Several flavonoids and polyphenolic compounds possess both antioxidant and anti-tyrosinase activities (Figure 1). Kim et al. [22] reported that the methanolic extracts of *P. frutescens* seeds exhibit dose-dependent inhibition of tyrosinase and radical scavenging against DPPH and ABTS free radicals. The potential anti-tyrosinase activity of *P. frutescens* seeds is attributed to phenolic compounds, such as rosmarinic acid, rosmarinic acid-3-*O*-glucoside, luteolin, apigenin, chrysoeriol, and caffeic acid. Rosmarinic acid exhibited potential anti-tyrosinase activity, with an IC_50_ value of 20.8 µM, followed by luteolin (24.6 µM), chrysoeriol (35.8 µM), apigenin (49.3 µM), rosmarinic acid-3-*O*-glucoside (57.9 µM), and caffeic acid (>300 µM). Furthermore, all of these compounds showed significant antioxidant activity, as evidenced by their DPPH and ABTS radical-scavenging capacities. Based on these activities, *P. frutescens* seeds can be used as health additives in food preparations.

### 3.5. Antispasmodic Effect

Vicenin-II, a bis C-glycosylflavonoid, displayed no direct spasmolytic effect, but significantly reduced Ba2+-or acetylcholine-induced smooth muscle contractions in the rat ileum (a part of the small intestine of the gastrointestinal tract), demonstrating an antispasmodic effect. The antispasmodic effect on smooth muscles relieves gastrointestinal discomfort and bowel disease symptoms, thus maintaining gut health. Hence, vicenin-II can be, prophylactically, used in maintaining and improving gut health [40].

### 3.6. Insecticidal Activity

Different active constituents reported in the essential oil, such as, R-(+)-carvone, perilla aldehyde, limonene, perillic acid, caryophyllene oxide, methyl perillate, perilla alcohol, and 2-furyl methyl ketone, obtained from *P. frutescens* aerial-dried parts, exhibited stronger insecticide and-repellent activities, against different insects. For example, 2-Furyl methyl ketone (LD_50_ = 0.86 mg/L air) exhibited stronger fumigant toxicity against *Lasioderma serricorne*, followed by R-(+)-carvone(LD_50_ = 1.83 mg/L air), perilla aldehyde (LD_50_ = 3.03 mg/L air), and the crude essential oil (LD_50_ = 4.16 mg/L air) of *P. frutescens*. The 2-Furyl methyl ketone, also, exhibited strong insecticidal activity against *Tribolium castaneum*, with an LD_50_ of 1.32 mg/L air (fumigant toxicity). Limonene, also, exhibited strong insecticidal activity against both *Lasioderma serricorne* and *Tribolium castaneum*, with LD_50_ values of 14.07 mg/L air and 6.21 mg/L air, respectively [7]. Limonene, perilla aldehyde, perillic acid, caryophyllene oxide, methyl perillate, and perilla alcohol exhibit stronger larvicidal activity against dengue-fever-causing mosquitoes, *Aedes aegypti* [5] Methyl perillate was the most toxic against *Aedes aegypti*, with an LC_50_ value of 16.0 ppm, followed by limonene (LC_50_ of 29.1 ppm), caryophyllene oxide (LC_50_ of 29.8 ppm), perilla aldehyde (LC_50_ of 35.3 ppm), perilla alcohol (LC_50_ of 39.1 ppm), and perillic acid (LC_50_ of 56.5 ppm). A sesquiterpenoid, α-farnesene, isolated from the whole plant extract of *P. frutescens*, exhibited insecticidal activity against third instar larvae of *Plutella xylostella*, with an LD_50_ of 53.7 ppm [72]. The results indicate that the essential oil of *P. frutescens* and isolated compounds have the potential to be developed into natural repellents or insecticides, for controlling insects in stored products. The development of natural pesticides would help to decrease the negative effects, such as pesticide residues, resistance, and environmental pollution, caused by synthetic insecticides and repellents.

### 3.7. Anti-Allergic Activity

Different bioactive components of *P. frutescens*, including rosmarinic acid, caffeic acid, luteolin, apigenin, methoxyflavanone and α-linolenic acid, were found to possess antiallegic activity (Figure 2) [73]. Kamei et al. [42] reported the presence of 8-hydroxy-5,7-dimethoxyflavanone in *P. frutescens* leaves, and named it Perilla-derived methoxyflavanone (PDMF). PDMF may be used to prevent IgE-driven type I hypersensitivity reactions. Akt phosphorylation and intracellular Ca^2+^ influx, the two critical molecular events involved in mast cell degranulation in allergic reactions, are suppressed by PDMF, which acts as a potent anti-allergic compound. The other polyphenols of *P. frutescens*, such as rosmarinic acid, luteolin, apigenin, and caffeic acid, also, suppress IgE-mediated type I hypersensitivity reactions. However, PDMF exhibits a more potent histamine-release inhibitory activity than known derived anti-inflammatory polyphenols. Further PDMF stimulation suppresses histamine release from RBL-2H3 cells in a dose-dependent manner, and the IC_50_ value (68.5 mM) was found to be considerably lower than that of apigenin (96.8 mM), luteolin (174.1 mM), caffeic acid (620.4 mM), and rosmarinic acid (>1000 mM).

Specific matrix proteins, such as matrix metalloproteinase (MMP), periostin, interleukin (IL-31), thymus, and activation-regulated chemokine (TARC), are sensitive markers for the clinical diagnosis of skin health [74]. These protein levels are drastically elevated during conditions of skin damage, such as skin injuries, skin inflammation, skin allergy, and atopic dermatitis; atopic dermatitis or atopic eczema is a type of severe skin inflammation, with symptoms such as swelling, redness, cracking, and itchy skin. Heo et al. [75] reported that an aqueous extract of *P. frutescens* mitigates DNFB (2,4-dinitrofluorobenzene)-induced atopic dermatitis, in a mouse model. The anti-dermatitis activity of the aqueous extract of *P. frutescens* is, possibly, due to the downregulation of matrix metalloproteinase-9 (MMP-9) and IL-31 expression levels, and upregulation of T-bet activity. Komatsu et al. [76] demonstrated that *P. frutescens* leaf extract prevents atopic dermatitis induced by house dust mite allergens (*Dermatophagoides farina*), in an NC/Nga AD mouse model. Moreover, *P. frutescens* leaf extract significantly decreased the serum levels of inflammatory markers, such as IgE, periostin, and TARC. Furthermore, *P. frutescens* leaf extract significantly decreased the allergen-stimulated CD4^+^/CD8^+^ T cell ratio, in spleen lymphocytes.

Shin et al. [77] demonstrated that an aqueous extract of *P. frutescens* effectively inhibited immediate mast-cell-triggered allergy and inhibited the fatal systemic allergy, using compound 48/80 (a potent inducer of systemic allergy, by stimulating mast cell degranulation and promoting histamine release), in a dose-dependent manner. Furthermore, they reported that an aqueous extract of *P. frutescens* significantly inhibited the local allergic reaction, induced by anti-DNP IgE. The anti-allergic effects of the aqueous extract of *P. frutescens* may be due to the inhibition of histamine released from mast cells, TNF-α production, and passive cutaneous anaphylaxis (PCA). Jeon et al. [78] reported that luteolin obtained from the methanolic extracts of *P. frutescens* exhibited potential antiallergic and antipruritic effects. The antiallergic effect of luteolin is due to the inhibition of histamine release from mast cells, induced by compound 48/80. Luteolin significantly inhibited the scratching behaviour and vascular permeability induced by serotonin and compound 48/80, respectively, in an ICR mouse model. Hyaluronidase is one of the major enzymes regulating mast cell degranulation, and histamine release is, primarily, controlled by protein kinase C and Ca^+2^. Asada et al. [79] reported that a glycoprotein from *P. frutescens* hot-water extracts prevented the degranulation of mast cells, possibly due to inhibition of protein kinase C and hyaluronidase activity. Chen et al. [80] reported that the ethanolic extract of *P. frutescens* successfully exhibited an antiallergic effect in an ovalbumin (allergen)-sensitised asthma mouse model. The ethanolic extract suppressed serum IgE levels, downregulated allergen-stimulated Th2 cytokines (IL-5 and IL-13), and decreased the secretion of allergic mediators (histamine and eotaxin). Furthermore, the ethanolic extract of *P. frutescens* inhibits airway hyper-responsiveness, by suppressing cell infiltration and reducing lung and bronchiole inflammation. Sanbongi et al. [81] reported that oral administration of rosmarinic acid extract of *P. frutescens* inhibits allergic inflammation, caused by the mite allergen *D. farina*. *D. farina*-sensitised C3H/He mice exhibited severe eosinophilic inflammation and elevated expression levels of IL-4, IL-5, and eotaxin in the lungs; rosmarinic acid extract significantly decreased the influx of eosinophils in the lungs and inhibited the lung tissue expression of IL-4, IL-5, and eotaxin proteins. In another study, by Takano et al. [82] rosmarinic acid-rich *P. frutescens* extract inhibited allergic rhinoconjunctivitis in humans, at least partly by inhibiting the infiltration of polymorphonuclear leukocytes (PMNL) into the nostrils. In this study, rosmarinic acid-rich extract significantly decreased the number of PMNL and inhibited the levels of inflammatory mediators, such as histamine, eotaxin, IL1-β, IL-8, and IgE levels, in nasal lavage fluid. Makino et al. [83] demonstrated that hot-water extracts of *P. frutescens* significantly suppressed the mice ear–PCA reaction, thereby concluding that rosmarinic acid is the active constituent, mainly, responsible for anti-allergic reactions. Dietary perilla oil, rich in α-linolenic acid, significantly suppressed serum lipid levels as well as IgG1 and IgA levels, in ovalbumin-sensitised mice. Based on these results, Chang et al. [84] reported that dietary perilla oil might, moderately, suppress asthmatic allergy. *P. frutescens*, PDMF, and rosmarinic acid could be, medially, useful in treating allergic diseases.

### 3.8. Anti-Depressant Activity

Figure 2 represents the different bioactive components of *P. frutescens* exhibiting antidepressant activity. Rosmarinic acid and caffeic acid showed significant antidepressant activity; this was demonstrated by Takeda et al. [85] in a forced swimming test in mice, which is a well-accepted stress model of depression. The antidepressant activity was revealed via mechanisms other than the inhibition of monoamine transporters and monoamine oxidase. In neuropharmacological studies, neither rosmarinic acid nor caffeic acid affected either the uptake of monoamines to synaptosomes or mitochondrial monoamine oxidase activity in the mouse brain. The detailed mechanisms involved in the antidepressive-like properties of rosmarinic acid and caffeic acids are unclear. However, previous pharmacological studies have revealed that rosmarinic acid inhibits histamine release from mast cells [86], as well as that caffeic acid can activate the a1-adrenoreceptor system [87] and inhibit the production and release of nitric oxide (NO) [88,89]. Furthermore, previous studies using the forced swimming test have suggested that either the activation of α_1_-adrenoreceptors or the inhibition of NO production may be involved in the expression of antidepressive-like effects. Their effects may, reportedly, involve direct modulation of a second messenger system. Caffeic acid, reportedly, inhibits both protein kinase A and protein kinase C activity in vitro. Increased evidence suggests that the therapeutic effects of existing antidepressants are associated with adaptive changes in post-receptor signalling, rather than with their primary action. Furthermore, EOPF exhibited significant antidepressant activity in a chronic, unpredictable, mild stress (CUMS)-induced mouse model, a widely used rodent model of depression. Brain-derived neurotrophic factor (BDNF) is a nerve growth factor essential for neuronal survival, and promotes the growth, differentiation, and maintenance of neurons, as well as synaptic plasticity, cognitive function, and long-term memory. BDNF is present throughout the brain and spinal cord, and its levels are decreased in conditions of depression, aging, and neurological disorders, including Alzheimer’s disease, Parkinson’s disease, Huntington’s disease, and Lou Gehrig’s disease. However, the hippocampal region of the brain shows high densities of BDNF. CUMS significantly decreased hippocampal BDNF protein levels, by downregulating the mRNA expression of BDNF. Administration of EOPF significantly elevates BDNF protein levels, by upregulating BDNF mRNA expression levels [90]. Thus, EOPF showed antidepressant activity, by enhancing BDNF protein levels. Chronic stress significantly decreases serotonin concentrations (5-hydroxytryptamine) and its metabolite 5-hydroxyindoleacetic acid (5-HIAA), in the hippocampus of the brain. Furthermore, chronic mild stress could induce a proinflammatory response, by increasing the plasma levels of pro-inflammatory cytokines, including IL-1, IL-6, and TNF-α. CUMS also reduced open-field activity and sucrose consumption as well as increased the duration of immobility in the forced swimming and tail suspension tests. Administering EOPF effectively increased the concentrations of serotonin and 5-HIAA and reduced the levels of IL-6, IL-1β, and TNF-α. Furthermore, EOPF significantly enhanced open-field activity and sucrose consumption as well as reduced the duration of immobility. Ji et al. [91] demonstrated that perillaldehyde, a major constituent of EOPF, exhibited an antidepressant effect in an LPS-induced depression mouse model. In this experiment, perillaldehyde elevated the reduced levels of monoamines, such as norepinephrine and 5-hydroxytryptamine, in the prefrontal cortex of LPS-depressed mice. Furthermore, perillaldehyde decreased proinflammatory cytokine levels (IL-6 and TNF-α) in the prefrontal cortex of mice, and significantly reduced the LPS-enhanced immobility duration in the forced swimming and tail suspension tests.

Luteolin and apigenin, isolated from the fruit of *Perilla frutescens* (L.) Britton enhance monoamine uptake, either on monoamine-transporter transgenic Chinese hamster ovary (CHO) cells or on wild dopaminergic cell lines, with higher specificity for dopamine (DA) uptake than for norepinephrine (NE) and serotonin (5HT) uptake, with a greater potency and efficacy for luteolin than for apigenin. Furthermore, in the transgenic cells, the principal NE/DA uptake activated by luteolin was significantly prevented by the respective transporter inhibitor, and the transmitter-uptake-enhancing action was independent of its ligands, which supports the compounds as monoamine transporter activators. Furthermore, luteolin, markedly, inhibited the cocaine-targeted effect in CHO cells overexpressing the dopamine transporter. Thus, luteolin and apigenin function as monoamine transporter activators, which would improve several hypermonoaminergicneuropsychological disorders, especially cocaine dependence, by upregulating monoamine transporter activity [92].

Luteolin and apigenin, both isolated from FP, are novel monoamine transporter activators, and luteolin is the most potent DAT activator, which, strikingly, prevents in vitro cocaine-targeted action. Luteolin, therefore, could be effective for hyperfunctional monoamine-associated neuropsychological disorders (e.g., mania, schizophrenia, and drug problems, especially cocaine dependence). The mechanism of the enhancement of DAT/NET function is unknown and needs to be determined in future studies [93].

### 3.9. Hepatoprotective Activity

Rosmarinic acid, a major polyphenolic component of *P. frutescens*, reduces lipopolysaccharide-induced liver injury in d-galactosamine-sensitised mice. The mechanism of action of perilla-derived rosmarinic acid, in reducing liver injury induced by DGalN and LPS, is, mainly, attributed to the scavenging of superoxide or peroxynitirite. Furthermore, rosmarinic acid decreases the mRNA expression levels of the proinflammatory cytokine TNF-α, a prime cause of liver injury. Additionally, it decreases the mRNA expression levels of inducible nitric oxide synthase (iNOS), an important enzyme that produces nitric oxide (NO), which plays an important role in the apoptosis of injured hepatocytes [93].

Caffeic acid and rosmarinic acid from *P. frutescens* leaves exhibit significant hepatoprotection against tert-butylhydroperoxide (t-BHP)-induced oxidative stress damage in the liver and suppress oxidative stress damage in different ways. They increase the levels of intracellular GSH, by enhancing the activity of γ-GCS, and enhance the activity of various antioxidant enzymes, such as SOD, GPx, and CAT. They, further, reduce the levels of oxidative stress biomarker enzymes, such as aspartate transaminase (AST), alanine transaminase (ALT), lactate dehydrogenase (LDH), and lipid peroxidation. Their combination showed synergistic hepatoprotective activity, compared to their individual action [94].

### 3.10. Hair Growth Promotion Activity

The butanolic fraction of *P. frutescens* extract (BFPE) promoted hair regrowth in C57BL/6 mice; BFPE successfully induced the anagen phase of hair growth in mice seven days after shaving. Furthermore, BFPE stimulated hair elongation through the proliferation and differentiation of hair follicles and prevented androgenic alopecia. The hair growth-promoting activity of BFPE is, mainly, due to its active constituent, rosmarinic acid, which increased the viability of PHFC (primary hair follicle fibroblast) cells. Furthermore, rosmarinic acid reduced testosterone- and dihydrotestosterone-induced androgenic alopecia. The antimicrobial, anti-inflammatory, anti-leukotriene B4, and anti-androgenic alopecia activities of rosmarinic acid in *P. frutescens* leaves, synergistically, promote hair growth [95].

### 3.11. Hypolipidemic Activity

The total flavonoid extract of *P. frutescens* (TFP), mainly, contains apigenin and luteolin. TFP inhibited hyperlipidemia in rats fed with a high-fat diet. TFP decreases lipid accumulation in adipose tissues and serum levels of triacylglycerols, total cholesterol, and low-density lipoprotein cholesterol (bad cholesterol), accompanied by increased levels of high-density lipoprotein cholesterol (good cholesterol). Furthermore, TFP suppressed oxidative stress in hyperlipidemic rats, by increasing the levels of antioxidant enzymes, such as SOD and GPx, following the inhibition of lipid peroxidation, by decreasing serum malondialdehyde (MDA) levels in the serum of high-fat-diet-fed rats. Based on this evidence, *P. frutescens* can be used as a food additive to prevent atherosclerosis [96].

### 3.12. Inotropic and Lusitropic Effects

Korotkich et al. [97] reported that *P. frutescens* extract (PFE) exhibited positive inotropic and lusitropic effects on the myocardium of rabbits. PFE increased myocardial contraction (inotropic) and relaxation (lusitropic) effects in a dose-dependent manner and demonstrated that the inotropic and lusitropic effects were attributed to the metabolism of calcium in cardiac muscle cells. An increase in the influx of calcium ions, through L-type membrane channels, increases the velocity of contraction (inotropic), and the acceleration of the sarcoplasmic reticulum uptake of calcium ions increases the relaxation velocity (lusitropic).

### 3.13. Neuroprotective Activity

Luteolin from the ripe seed of *P. frutescens* exhibits neuroprotective effects against ROS-induced cytotoxicity in primary cortical neurons. Luteolin enhanced the viability of H_2_O_2_ intoxicated primary neurons [98] inhibited ROS production in primary neurons treated with H_2_O_2_, and enhanced mitochondrial membrane potential in a concentration-dependent manner. Furthermore, luteolin decreased oxidative stress in primary cortical neurons, by enhancing the levels of antioxidant enzymes, such as CAT and GSH. Hence, luteolin can be used as a dietary supplement to prevent neurodegenerative disorders such as Alzheimer’s, Parkinson’s, and cerebral ischemia disorders. Kim et al. [99] reported that luteolin, from the alcoholic extract of *P. frutescens*, inhibited NO production in microglial cells, by suppressing the mRNA expression levels of iNOS. Rosmarinic acid from the methanolic extract of *P. frutescens* exhibited neuroprotective effects, against oxidative stress induced by H_2_O_2_ in C6 glial cells [100]. H_2_O_2_-intoxicated glial cells showed elevated levels of ROS, NO production, and lipid peroxidation, which are associated with apoptosis of glial cells. H_2_O_2_ intoxication upregulated the mRNA expression levels of iNOS and COX-2. Rosmarinic acid treatment protects glial cells, by reducing ROS, NO, and lipid peroxidation. The neuroprotective effect of rosmarinic acid was due to the downregulation of protein and mRNA expression levels of iNOS and COX-2. Lee et al. [101] demonstrated that the methanolic extract of *P. frutescens* and its active constituent rosmarinic acid effectively improved cognitive function and exhibited objective discrimination in an amyloid β (Aβ_25–35_)-induced mouse model of Alzheimer’s disease (AD). The methanolic extract of *P. frutescens* and rosmarinic acid inhibited the production of NO (neurotoxicant) and MDA (causing cell membrane damage by lipid peroxidation) in Aβ_25–35_-injected mouse brain. Perilla seed oil and its active constituent α-linolenic acid (ALA) protect SH-SY5Y (human neuroblastoma) cells from H_2_O_2_-induced oxidative stress and apoptosis; the cell death of SH-SY5Y cells is attenuated, by downregulating the Bax/Bcl-2 ratio, cleaved PARP, cleaved caspase-3, and caspase-9 (Lee et al., 2018). β-site amyloid precursor protein cleaving enzyme 1 (BACE 1) is one of the three enzymes required for the production of β-amyloid, which is a neurotoxic peptide that plays a major role in Alzheimer’s disease. Inhibition of BACE 1 limits or stops the production of β-amyloid, which, consequently, slows the pathology of Alzheimer’s disease. Choi et al. [102] reported that luteolin and rosmarinic acid from methanolic extracts of *P. frutescens* inhibited the BACE 1 enzyme activity, in a dose-dependent manner, with IC_50_ values of 0.5 and 21 µM, respectively, and inhibited the BACE 1 enzyme in a non-competitive manner. Perilla seed oil and leaf extract showed neuroprotective effects against β-amyloid-peptide-induced toxicity in pheochromocytoma (PC12) cells, reducing oxidative stress and inhibiting hyperphosphorylation of tau protein. Tau protein is a microtubule-associated protein that maintains the normal shape and structure of neurons, which is hyperphosphorylated in neurodegenerative diseases and is incapable of maintaining the normal neuron morphology. Perilla seed oil and leaf extract successfully inhibited tau protein phosphorylation and enhanced neurite-outgrowth-bearing cells [103].

### 3.14. Anti-Inflammatory Activity

More than ten different bioactive compounds of *P. frutescens* showed significant anti-inflammatory activity (Figure 3). Wang et al. [8] reported that the aerial parts of *P. frutescens* contain seven monoterpenoids, which include three new furanoid monoterpenoids, frutescenone A (furanoid with 2,3-bifuran skeleton), frutescenone B (8R-hydroxyperillaketone), frutescenone C (perillaketone-adenine hybrid heterodimer), isoegomaketone, 9-hydroxyisoegomaketone (perillaketone monoterpenoids), (3S,4R)-3-hydroxyperillaldehyde, and (S)-(−)-perillic acid; six phenylpropanoids, such as methylisoeugenol, elemicin, isoelemicin, 3,4,5-trimethoxycinnamyl alcohol, myristicin, and ethyl caffeic acid; and three alkaloids, indole-3-carboxaldehyde, 1H-indole-3-carboxylicacid, and neoechinulin A. These monoterpenoids, phenylpropanoids, and alkaloids exhibit anti-inflammatory effects, by inhibiting the production of proinflammatory cytokines (TNF-α and IL-6) and proinflammatory mediators (NO) in LPS-stimulated RAW264.7 cells. Two neolignans, magnosalin and amanicin, have the potential to treat endotoxemia and inflammation, accompanied by the overproduction of NO and TNF-α.

The inflammatory mediator NO is a marker that evaluates the anti-inflammatory effect of a drug or compound and is synthesised by the inducible NO synthase (iNOS) enzyme. The iNOS gene expression is induced by the proinflammatory cytokine IL-1β. Different compounds, such as luteolin, apigenin, negletein, shisoflavanone A (8-hydroxy-6,7-dimethoxyflavanone), 5,8-dihydroxy-7-methoxyflavanone, esculetin, and protocatechuic acid, exhibit significant anti-inflammatory activity, by decreasing IL-1β-induced NO production [28]. These compounds, further, decrease the mRNA expression levels of iNOS and TNF-α and suppress NO production in a dose-dependent manner. Shisoflavanone A, apigenin, and negletein exhibited strong NO suppression activity, with IC_50_ values of 10 µM, 12 µM, and 15 µM, respectively. Esculetin, luteolin, and 5,8-dihydroxy-7-methoxyflavanone, also, suppressed NO production, with IC_50_ values of 34 µM, 39 µM, and 55 µM, respectively.

Huang et al. [104] reported that the methanolic leaf extract of *P. frutescens* exhibited anti-inflammatory activity in lipopolysaccharide (LPS)-stimulated macrophage (RAW264.7) cells, which ameliorates LPS-induced inflammation, by downregulating the mRNA expression levels of proinflammatory markers (IL-2, IL-6, and TNF-α) and inflammatory mediators (iNOS and COX-2). Their study, also, elucidated the underlying mechanism of inhibition of proinflammatory markers and inflammatory mediators. Phosphorylation of MAPKs (ERK1/2, JNK, and p38) and nuclear translocation of NF-κB are the two main factors associated with the production and expression of proinflammatory cytokines and inflammatory mediators. *P. frutescens* leaf extract inhibits MAPK phosphorylation, nuclear translocation of NF-κB, and cytosolic IkBα degradation, in LPS-stimulated macrophages. Chang et al. [84] reported that *P. frutescens* oil, rich in α-linolenic acid, suppresses bronchoalveolar inflammation in an ovalbumin-sensitised-asthmatic-mouse model. *P. frutescens* oil suppresses the levels of proinflammatory markers, such as IL-1β, IL-2, IL-4, IL-5, IL-6, and IL-10.

Triterpene acids from *P. frutescens* leaves inhibit ear oedema inflammation, induced by tetradecanoyl phorbol acetate (TPA), in ICR female mice. Tormentic acid exhibits strong anti-inflammatory effect, with an IC_50_ value of 0.03 mg/ear, followed by corosolic acid and augustic acid (IC_50_ of 0.09 mg), ursolic acid and 3-epimaslinic acid (IC_50_ of 0.10 mg), pomolic acid (IC_50_ of 0.12 mg), hyptadienic acid (IC_50_ of 0.13 mg), and oleanolic acid (IC_50_ of 0.3 mg) [14]. The ethanolic extract of *P. frutescens* leaves (PFLE) showed anti-inflammatory effects on human neutrophils, stimulated by N-formyl-Met-Leu-Phe (fMLF), a potent macrophage activator and PMNL chemotactic factor (Chen et al., 2015). The underlying mechanisms revealed that PFLE exhibited dose-dependent inhibition of elastase release, ROS formation, superoxide anion production, cell migration, and CD11b expression. Furthermore, PFLE inhibits the activation of Src family kinases, including Src (Tyr416) and Lyn (Tyr396), and decreased intracellular Ca^+2^ mobilization. Urushima et al. [105] demonstrated that *P. frutescens* extract suppressed inflammatory bowel disease, induced by dextran sulfate sodium in a C57/BL6 mouse model, and reported that *P. frutescens* extract inhibits excessive secretion of proinflammatory cytokines (TNF-α, IL-1β, IL-6, and IL-17A) in the colon and upregulates anti-inflammatory cytokines (IL-10 and TGF-β). Immunocompetent cells, such as CD4^+^Foxp3^+^Tregs, regulate the pathogenesis of ulcerative colitis, and, mainly, induce the secretion of anti-inflammatory cytokines. Furthermore, Urushima et al. [105] elucidated that the prepared extract, mainly, comprises three compounds, luteolin, apigenin, and rosmarinic acid, wherein luteolin suppressed the production of TNF-α, IL-1β, IL-6, and IL-17A; apigenin suppressed IL-1β and IL-17A levels; and rosmarinic acid significantly reduced the levels of IL-1β and enhanced the levels of IL-10 and TGF-β, by upregulating the mRNA expression levels of Tregs. Apigenin enhanced the levels of IL-10 levels through other mechanisms; however, luteolin could not induce IL-10 production. Jeon et al. [78] reported that luteolin from *P. frutescens* methanolic extracts exhibited potential anti-inflammatory effects. Luteolin inhibited the production of TNF-α and IL-1β in human mast cells, induced by PMA and Ca^+2^ ionophore.

Isoegomaketone, isolated from *P. frutescens* essential oil, exhibited anti-inflammatory effects in LPS-induced mouse macrophages (RAW 264.7 cells) and suppressed the production of NO (inflammatory mediators), IL-6 (inflammatory cytokine), and monocyte chemoattractant type-1 (MCP-1, a protein that attracts monocytes and macrophages to the areas of inflammation). Park et al. [54] prepared different synthetic derivatives of isoegomaketone. One of the isoegomaketone derivatives inhibited NO, MCP-1, and IL-6 more effectively than its precursor. Park et al. [54] also, demonstrated that isoegomaketone and its derivatives suppressed the NF-kB pathway and activator protein-1 (AP-1), which are two inflammation-associated genes involved in the production of inflammatory mediators and inflammatory cytokines. Moreover, 9-hydroxy-isoegomaketone from *P. frutescens* leaves inhibited NO production in the LPS-stimulated mouse macrophage cell line RAW 264.7, as 9-hydroxy-isoegomaketone inhibits NO production, in a dose-dependent manner, with an IC_50_ value of 14.4 µM [55].

Yang et al. [20] reported that rosmarinic acid from *P. frutescens* leaves showed anti-inflammatory effects in cecal ligation and puncture (CLP)-induced septic mice and in human endothelial cells. Rosmarinic acid suppresses phorbol myristateacetate (PMA), TNF-α, IL-1β, and CLP-mediated endothelial protein C receptor (EPCR) shedding, by inhibiting the expression of TNF-α and converting enzyme (TACE). Furthermore, rosmarinic acid suppressed the phosphorylation of p38, ERK1/2, and JNK stimulated by PMA [106]. High-mobility group box-1 (HMGB-1) is a DNA-binding nuclear protein that acts as a proinflammatory cytokine, which plays an important role in infections, injury, and sepsis-related inflammation. Rosmarinic acid suppressed CLP-or LPS-stimulated HMGB1 release, and inhibited HMGB1 mediated inflammation in endothelial cells and sepsis-related mortality. Based on the above studies, rosmarinic acid can be used as a potential therapeutic agent for treating vascular inflammatory diseases, by inhibiting the HMGB-1 signalling pathway.

Luteolin reported from the *P. frutescens* alcoholic extracts inhibits neuro-inflammatory response in LPS-activated microglial (BV-2) cells, through the suppression of NO production [99]. Luteolin suppressed NO production in a dose-dependent manner, with an IC_50_ value of 6.9 µM, and inhibited NO production by suppressing the mRNA expression levels of iNOS. Deep-mechanism studies demonstrated that luteolin suppressed the degradation of Ik-Bα, an inhibitor of NF-kB, which is a protein family that mediates inflammatory responses, by producing inflammatory mediators and proinflammatory cytokines. Thus, luteolin inhibits the neurotoxic effector (NO) in the central nervous system, by suppressing the mRNA expression levels of iNOS [99].

The essential oil of *P. frutescens* (EOPF) suppressed chronic stress-induced neuro-inflammatory responses, by inhibiting the plasma levels of proinflammatory cytokines, including IL-1, IL-6, and TNF-α. Ji et al. [91] reported that perillaldehyde, a major constituent of EOPF, could reduce neuroinflammation by inhibiting IL-6 and TNF-α levels in the prefrontal cortex of an LPS-induced mouse model. Reflux of gastric contents from the stomach into the oesophagus causes inflammation of mucosal tissue and oxidative stress, a condition called reflux oesophagitis. Arya et al. [107] demonstrated that *P. frutescens* fixed oil successfully inhibited experimental reflux oesophagitis in Wistar albino rats. In this experiment, *P. frutescens* fixed oil exhibited different activities, such as antioxidant, lipoxygenase-inhibitory, anticholinergic, and antihistamine activities. These activities of *P. frutescens* fixed oil are, mainly, due to the presence of α-linolenic acid. Together, these activities provide protection against reflux oesophagitis in experimental rats.

### 3.15. Antioxidant Activity

Oxidative stress is, mainly, referred to as an imbalance between reactive oxygen species (free radicals) and antioxidant capacity, occurring when ROS production exceeds antioxidant capacity. It causes severe cellular damage, by disrupting cellular defence systems by interfering with cellular survival mechanisms, and is associated with several human diseases, including aging, neurodegenerative, cardiovascular, carcinogenesis, hepatic, and kidney diseases [108]. Oxidative stress-induced cell death can be weakened, by suppressing intracellular ROS production and by inducing the expression of antioxidant enzymes and cytoprotective proteins, such as superoxide dismutase (SOD), glutathione peroxidase (GP_X_), thioredoxin, catalase, γ-glutamylcysteine synthetase (g-GCS), glutathione S-transferase (GST), hemeoxygenase-1 (HO-1), and NAD(P)H: quinone oxidoreductase-1 (NQO1). Moreover, the nuclear factor erythroid 2-related factor 2 (Nrf2)-antioxidant response element (ARE) pathway controls the expression of these proteins and enzymes, and its activation protects cells against oxidative stress, thus functioning as a cellular defence system [109].

In addition, 2^1^,3^1^-dihydroxy-4^1^,6^1^-dimethoxychalcone (DDC), a flavonoid isolated from the leaves of green perilla, suppresses intracellular ROS production. Furthermore, it stimulates the expression of antioxidant enzymes, such as γ-GCS, NQO1, and HO-1. DDC increases glutathione content and antioxidant response element (ARE) activity in a concentration-dependent manner. Nrf2 induces the expression of antioxidant genes, by binding to AREs, which is crucial in the antioxidant genes expression, to overcome the cellular damage caused by oxidative stress. DDC significantly induced Nrf2-dependent ARE activation. Furthermore, phosphorylation of Nrf2 plays an important role in nuclear translocation and transcriptional activation, through AREs. The Nrf2-ARE pathway is regulated by different signalling kinases, including p38 MAPK (p38 mitogen-activated protein kinase), PI3K, and protein kinase C. DDC, effectively, increased the p38 MAPK and PI3K/Akt pathways. Thus, DDC from green Perilla acts as an effective antioxidant [43]. Figure 4 depicts a schematic representation of Nrf2-dependent ARE pathway activation, by the DDC of *P. frutescens*.

γ-Glutamylcysteine synthetase (γ-GCS) catalyses the major regulatory (rate-limiting) step in the de novo synthesis of glutathione (GSH), a major antioxidant tripeptide that protects cells from cellular damage caused by ROS. Park et al. [26] reported that caffeic acid from the leaves of green perilla increased γ-GCS activity in a dose-dependent manner. Similarly, GSH content increased, according to γ-GCS activity. This result revealed that the elevation of γ-GCS activity by caffeic acid is responsible for the de novo increase in GSH synthesis.

The ethanol extract of purple perilla leaves contains different phenolic acids, such as hydroxybenzoic acids, which include isovanillic acid, protocatechuic acid, gallic acid; and hydroxycinnamic acids, which include rosmarinic acid, caffeic acid, sinapic acid, ferulic acid, chlorogenic acid, rosmarinic acid-3-*O*-glucoside, caffeic acid-3-*O*-glucoside, and luteolin-5-*O*-glucoside, which exhibits effective in vitro antioxidant activity against ABTS, DPPH, BHT, and H_2_O_2_ radicals. Among these compounds, rosmarinic acid is a strong natural antioxidant. Rosmarinic acid and luteolin exhibit strong scavenging activities against DPPH free radicals, with IC_50_ values of 8.61 mM and 7.50 mM, respectively [24].

### 3.16. Anticancer Activity

The ethanolic leaf extract of *P. frutescens* (PFLE) inhibited the viability of HL-60 human leukemia cells, in a concentration-dependent manner. PFLE caused DNA defragmentation and arrested the cell cycle in the G1-phase. Its treatment increased the activities of apoptotic proteins, including caspase-3, caspase-8, and caspase-9, in a dose-dependent manner, and induced the cleavage of poly (ADP-ribose) protein (PARP), a DNA-repair enzyme. Furthermore, PFLE upregulated the levels of the proapoptotic protein Bax and downregulated the levels of the anti-apoptotic protein Bcl-2. PFLE treatment increased the levels of glucose-regulated protein-78 (GRP-78), phosphorylated eIF-2α, and JNK and p21 levels, in a concentration-dependent manner. Kwak et al. (2009) demonstrated that PFLE strongly induced the apoptosis of HL-60 through different mechanisms, including mitochondria-mediated (upregulation of caspase-9 by releasing cytochrome C from mitochondria), death-receptor-mediated (upregulation of key initiator caspase-8), and endoplasmic-reticulum-stress-mediated (upregulation of GRP78 and activation of eIF-2α and JNK by phosphorylation) pathways. The activity of PFLE towards apoptosis of HL-60 might be attributed to its active constituent luteolin.

*P. frutescens* leaf extract (PLE) inhibited hepatic carcinoma (Hep-G2) in a dose-dependent manner, with an IC_50_ value of 105 µg/mL, and exhibited antiproliferative activity in Hep-G2 cells, by inducing apoptosis in both caspase-dependent and caspase-independent pathways. PLE upregulated the mRNA expression levels of pro-apoptotic proteins caspase-8 and Bax, whereas the mRNA expression levels of the anti-apoptotic protein Bcl-2 were downregulated by PLE treatment. PLE upregulated apoptotic inducers, including c-Jun, Jun-B, and Fos-B, in the FOS/JUN-pathway, and NFkBIA and TNFSF9, through the autocrine pathway. The apoptotic effect of PLE on Hep-G2 cells was attributed to its active constituents, such as rosmarinic acid, caffeic acid, luteolin, and triterpene acids [110].

Kwak et al. [111] demonstrated that the ethanol extract of perilla leaf (PLE) inhibited colorectal cancer (HCT116) and non-small cell lung cancer (H1299), in a dose-dependent manner, with IC_50_ values of 50 µg/mL and 67 µg/mL, respectively. The PLE-treated cells exhibited nuclear fragmentation and chromatin condensation. PLE treatment significantly enhanced the sub-G1 cell population and inhibited the migration of non-small cell lung cancer cells and their adhesion.

Akihisa et al. [15] reported nine pentacyclic triterpene acids from ethanolic extracts of *P. frutescens* leaves, including six ursane types (ursolic acid, corosolic acid, epicorosolic acid, pomolic acid, tormentic acid, and hyptadienic acid) and three oleanane types (oleanolic acid, augustic acid, and epimaslinic acid), and evaluated their anticancer activities against human cancer cell lines, including leukaemia (HL-60), breast cancer (MCF-7), and hepatic carcinoma (Hep-G2). Ursolic acid, corosolic acid, tormentic acid, oleanolic acid, and 3-epimaslinic acid showed dose-dependent cytotoxicity against all cancer cell lines. Ursolic acid, corosolic acid, and oleanolic acid exhibited significant cytotoxicity against leukaemia, breast, and hepatic carcinomas, with IC_50_ values between 12–48 µM. Tormentic acid and 3-epimaslinic acid exhibited moderate cytotoxicity, with IC_50_ values greater than 50 µM. Among the nine pentacyclic triterpene acids, ursolic acid, reportedly, exhibited strong anticancer activity against leukaemia, breast, and hepatic carcinomas, with IC_50_ values of 11.8 µM, 14.9 µM, and 27.6 µM, respectively.

Banno et al. [14] reported that lipophilic triterpene acids, such as tormentic acid and oleanolic acid from the ethanolic extracts of Perilla leaves, showed inhibitory activity against skin carcinoma, through the protein kinase C signalling pathway. Lipophilic pentacyclic triterpene acids, such as ursolic acid, oleanolic acid, maslinic acid, 3-epimaslinic acid, corosolic acid, and 3-epicorosolic acid, inhibit skin carcinoma promoted by 12-*O*-tetradecanoylphorbol-13-acetate (TPA) in an ICR mouse model. All pentacyclic triterpene acids inhibited the TPA-induced epidermal proliferation. Furthermore, they inhibited skin inflammation by suppressing the expression of inflammatory genes. However, maslinic acid and 3-epicorsolic acid were more effective against TPA-mediated skin tumour promotion. They inhibited epidermal proliferation and skin inflammation compared to other pentacyclic triterpene acids. Maslinic acid and 3-epicorsolic acid, effectively, inhibited skin carcinoma, by suppressing the expression of Cox-2 and Twist-1 proteins, and inhibited the phosphorylation (activation) of epidermal signalling pathway proteins, such as IGF-1βR, Src, STAT3, JNK1/2, and c-Jun. Furthermore, they enhanced the levels of the tumour-suppressor protein, Pdcd4 [80].

Seven anthocyanins reported from *P. frutescens* leaves, such as shisonin, cis-shisonin, malonyl-shisonin, malonyl-cis-shisonin (coumaroyl anthocyanins), two caffeoyl anthocyanins, and one feruloyl anthocyanin, exhibited significant apoptotic effects on human cervical adenocarcinoma (HeLa) cell lines, in a dose-dependent manner [44].

Isoegomaketone, from the aerial parts of green perilla, inhibited the viability of mouse melanoma (B16) cells, in a concentration-dependent manner. It inhibits melanoma growth in tumour-transplanted mice. The possible effects of the antitumor/anticancer activity of isoegomaketone are, also, explained. Isoegomaketone induces apoptosis of melanoma cells, via both caspase-dependent and caspase-independent pathways. It induces ROS generation, nuclear fragmentation, cell shrinkage, and cell membrane blebbing, and increases caspase-3 and caspase-9 activities, in a concentration-dependent manner. Isoegomaketone upregulates Bax (pro-apoptotic protein) and downregulates Bcl-2 (anti-apoptotic protein). Furthermore, isoegomaketone significantly increases Bax/Bcl-2 [9].

Dietary perilla oil inhibits methyl nitrosourea (MNU)-induced colon cancer in F344/NSlc rats. Narisawa et al. [112] demonstrated that perilla oil inhibits colonic tumour development by MNU, mainly because of the presence of α-linolenic acid, a ω-3 PUFA. α-Linolenic acid modified the sensitization of colonic cell membranes to carcinogens, such as MNU.

Perillaldehyde, a major volatile component of the essential oil of *P. frutescens*, inhibited the viability of mouse gastric cancer cell lines (MFCs). Perillaldehyde inhibits MFCs by inducing autophagy in MFCs, which is dependent on adenosine monophosphate-activated protein kinase (AMPK), and AMPK activation is, LKB1 dependent, a primary upstream kinase of AMPK. Zhang et al. [113] reported that PAH activates AMPK, via phosphorylation of the Thr172 residue. Phosphorylated AMPK induces autophagy in gastric cell lines. Furthermore, perillaldehyde upregulated the levels of apoptotic proteins, such as caspase-3 and p53, and promoted apoptosis of gastric cancer cells in vivo.

### 3.17. Antimicrobial Activity

The leaves of *P. frutescens* contain different volatile compounds such as terpenoids, which include perillaldehyde, β-caryophyllene, limonene, α-bergamotene, perillyl alcohol, isoeugenol, and linalool. Perillaldehyde showed broad-spectrum activity against different microbes, including Gram-negative bacteria (*Enterobacter aerogenes*, *Escherichia coli*, *Salmonella choleraesuis*, and *Pseudomonas aeruginosa*), Gram-positive bacteria (*Bacillus subtilis*, *Propionibacterium acnes*, *Staphylococcus aureus*, and *Streptococcus mutans*), and fungi and yeast (*Aspergillus niger*, *Candida albicans*, *Candida utilis*, *Penicillium chrysogenum*, *Mucor mucedo*, and *Saccharomyces cerevisiae*). Limonene and β-caryophyllene exhibited excellent antimicrobial activity against Gram-positive bacteria, especially *A. naeslundii* and *P. acnes*, respectively. β-Pinene, perillyl alcohol, isoeugenol, and linalool exhibited weak-butbroad-spectrum antimicrobial activity, against both Gram-positive bacteria and fungi (Kang et al., 1992) [114]. Luteolin, one of the phenolic compounds of Perilla seed, exhibited the strongest effect on the oral pathogenic bacterium *Porphyromonas gingivalis* (Yamamoto et al., 2002) [115].

Perilla essential oil showed antifungal activity against phytopathogenic fungi, including *A. niger*, *A. flavus*, *A. oryzae*, *Rhizopus oryzae*, and *Alternaria alternata* [3]. Rosmarinic acid, and its extracts from *P. frutescens* leaves, exhibited strong antimicrobial activity against different bacterial and fungal pathogens, including *E. coli*, *S. aureus*, *B. subtilis*, *A. niger* and *C. albicans*. Rosmarinic acid exhibits antibacterial activity, by targeting deformylase, which is a bacterial enzyme peptide that is essential in synthesising bacterial functional polypeptides. Rosmarinic acid shows fungal inhibitory activity, by targeting the fungal co/post-translational enzyme N-myristoyltransferase [116].

Among all compounds reported from *P. frutescens*, rosmarinic acid and luteolin were found to show a wide variety of biological applications. Figure 5 shows the different biological functions of the rosmarinic acid and luteolin of *P. frutescens*.

*Perilla**frutescens* shows no toxicity and no side effects to human. However, it shows few side effects for cattle and livestock. However, the side effects are negligible, when the cattle eat limited *P. frutescens*. *Perilla*
*frutescens* contains a pneumotoxin in the leaves and seeds that when metabolised in the rumen produces toxic intermediaries. *Perilla* ketone, from the essential oil of *Perilla* frutescens, is a pulmonary edemagenic agent for laboratory animals and livestock, when they take a high amount of *P. frutescens*.

## 4. Conclusions

*P. frutescens* is an annual herb, belonging to the family Lamiaceae, and is mainly cultivated in China, Japan, Korea, and India. *P. frutescens* has gained attention from researchers worldwide, owing to its emerging economic importance. The enormous phytochemical constituents, their medicinal properties, and their nutraceutical properties make *P. frutescens* a unique crop. The leaves, aerial parts, and seeds of *P. frutescens* are used for culinary and non-culinary purposes. Approximately 400 compounds, including alkaloids, terpenoids, quinines, phenylpropanoids, glycosides, benzoxipen derivatives, polyphenolic compounds, flavonoids, chalcones, coumarins, carotenoids, anthocyanins, neolignans, fatty acids, policosanols, tocopherols, and phytosterols, were reported from *P. frutescens*. Rosmarinic acid, luteolin, apigenin, caffeic acid, and their derivatives, which are present in leaves, seeds, and aerial parts, were the most active constituents of *P. frutescens*. Rosmarinic acid, luteolin, apigenin, and caffeic acid are the key bioactive constituents responsible for many biological activities, including antimicrobial, antioxidant, anticancer, antidepressant, anti-allergic, anti-inflammatory, antidiabetic, and hypolipidemic activities. Rosmarinic acid, luteolin, apigenin, and caffeic acid exhibited neuroprotective, hepatoprotective, and cardioprotective effects. Further Rosmarinic acid and luteolin exhibited xanthine oxidase, tyrosinase, alpha-glucosidase, hyaluronidase, and aldose reductase inhibitory activities. The antimicrobial, anti-inflammatory, anti-leukotriene B4, and anti-androgenic alopecia activities of rosmarinic acid and luteolin were, synergistically, involved in the hair growth promotion of *P. frutescens* leaf. Rosmarinic acid was found at very high concentrations in *P. frutescens*, compared to caffeic acid, luteolin, and apigenin. The essential oil of *P. frutescens* (EOPF) exhibited stronger insecticidal, antiplasmodial, antifungal, antibacterial, and antiviral activities, and demonstrated anticarcinogenic effects, in different cancer cell lines. EOPF suppresses the neuroinflammatory response, inhibits inflammation of the mucosal layer of the oesophagus, and exhibits anti-inflammatory effects. Furthermore, EOPF shows radical scavenging, lipoxygenase inhibitory, anticholinergic, and antihistamine activities. All of these EOPF activities were due to the presence of 200 different volatile substances, including monoterpenes, sesquiterpenes, and phenylpropanoids. Isoegomaketone and perillal were found to be the most active constituents of EOPF, with anti-inflammatory, insecticidal, antimicrobial, and anticarcinogenic effects. *P. frutescens* leaves constitute triterpene acids, anthocyanins, coumarins, carotenoids, and neolignans. Nine different types of triterpene acids present in *P. frutescens* exhibited anticarcinogenic effects against different types of human cancer cell lines. Seven different types of anthocyanins present in *P. frutescens* showed anti-inflammatory and anticancer activities. Coumarins and carotenoids present in *P. Frutescens* demonstrated xanthine-oxidase-inhibitory, anti-inflammatory, antioxidant, and anticancer activities. Two neolignans present in *P. frutescens* can, potentially, treat endotoxemia and inflammation. Perilla leaves are of industrial importance because of their medicinal and nutraceutical properties. Perilla leaves are edible as well as nutritious and are used as leafy vegetables and spicy herbs in Asian cuisine. The leaves are used in pickles, soups, salads, condiments, garnishes, food flavours, and wraps. Perilla seed oil, extracted from toasted seeds, is used for cooking in China, Japan, and Korea. Perilla seed oil is an important edible oil used for flavour enhancement, cooking, roasting, and dipping sauce. The seed oil is rich in polyunsaturated fatty acids, policosanols, phytosterols, and tocopherols. α-linolenic acid constitutes about 55–65% of the polyunsaturated fatty acids of *P. frutescens*. Perilla seed oil contains the highest proportion of omega-3-fatty acids, compared to any other plant oil worldwide. Perilla seed oil and α-linolenic acid exhibit strong neuroprotective effects, and tocopherols show strong antioxidant effects. Perilla seeds are, also, rich in flavonoids and polyphenolic compounds. Perilla seed oil, extracted from untoasted seeds, has been used for fueling lamps, printing ink, varnishes, paints, linoleum, lacquers, and waterproof coatings. The seed cake obtained after oil extraction has been used as an animal feed and a biofertiliser. Thus, in this review, we discuss the phytochemical constituents, biological properties, medicinal uses, and pharmaceutical and nutraceutical importance of *P. frutescens*. This review can serve as a ground work for subsequent studies on *P. frutescens*.

## Figures and Tables

**Figure 1 molecules-27-03578-f001:**
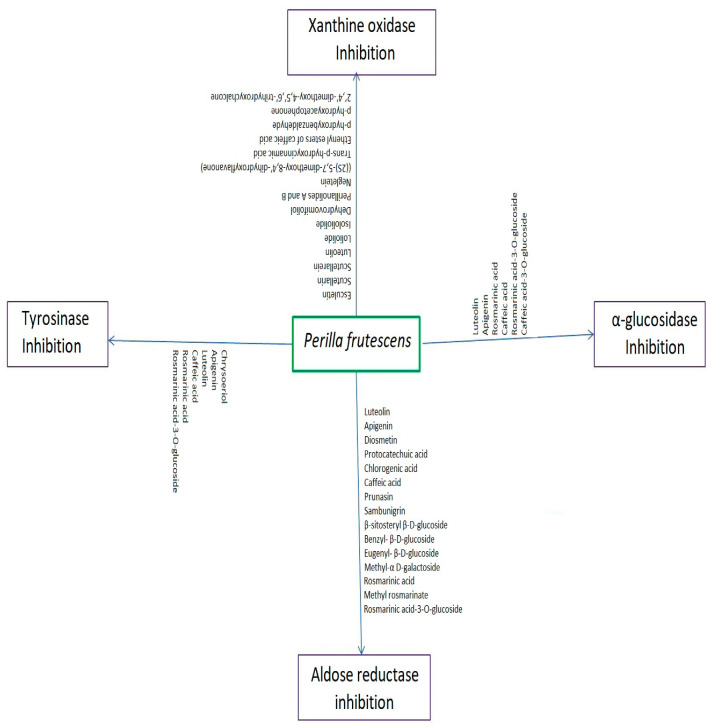
Different bioactive compounds of *P. frutescens* involved in enzyme inhibitory activities.

**Figure 2 molecules-27-03578-f002:**
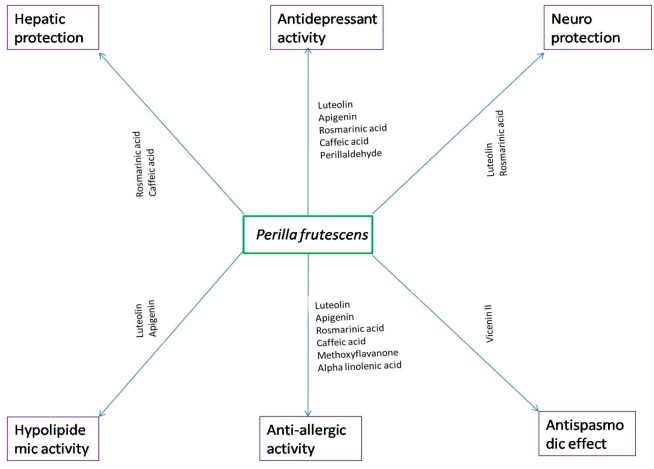
Different bioactive compounds of *P. frutescens*, exhibiting different biological functions, including antiallergic activity, antidepressant activity, antispasmodic effect, hypolipidemic, hepatoprotection, and neuroprotection activities.

**Figure 3 molecules-27-03578-f003:**
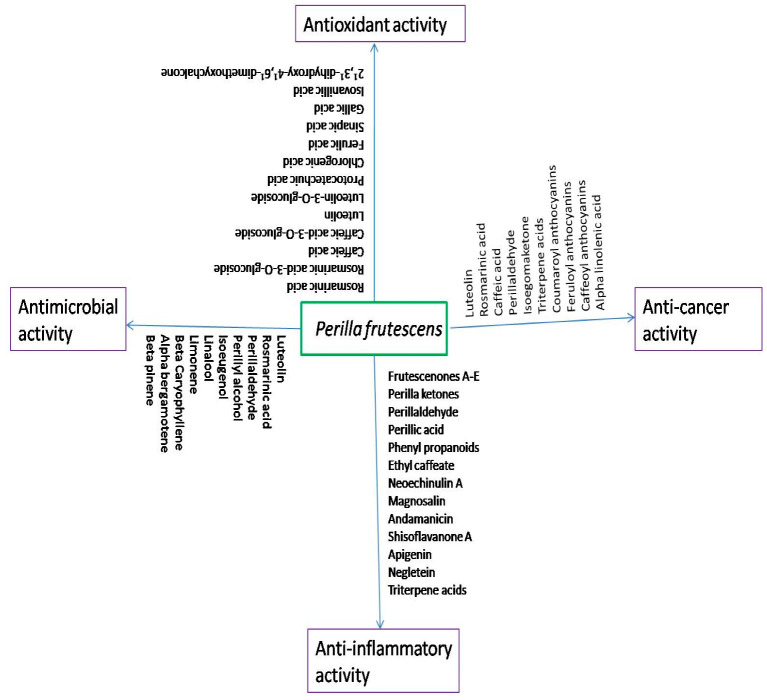
Various bioactive compounds of *P. frutescens*, exhibiting different biological functions, including antioxidant, anti-inflammatory, anticancer, and antimicrobial activities.

**Figure 4 molecules-27-03578-f004:**
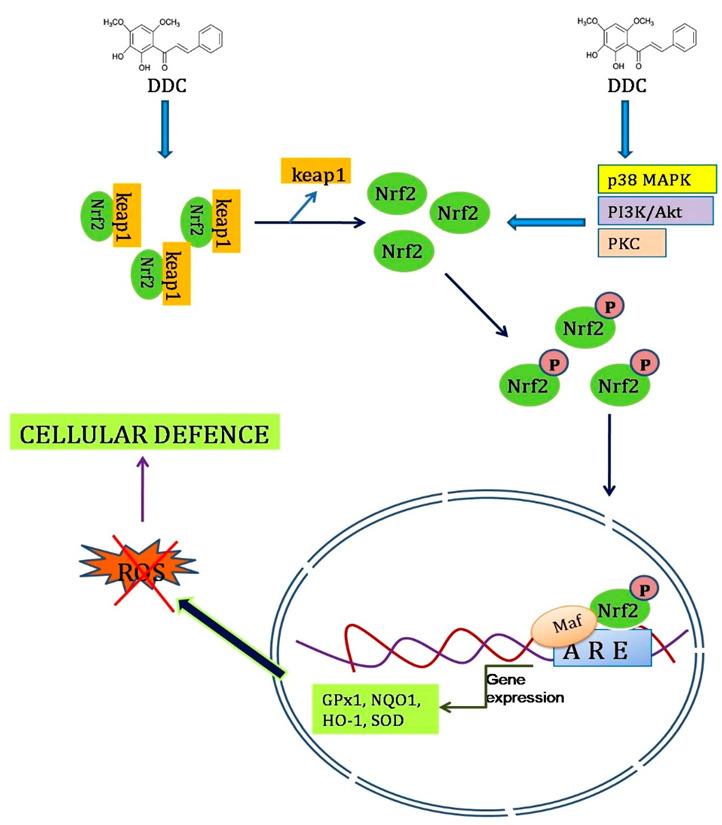
Schematic representation of Nrf2-ARE activation, by 2^1^,3^1^-dihydroxy-4^1^,6^1^-dimethoxychalcone from *P. frutescens*.

**Figure 5 molecules-27-03578-f005:**
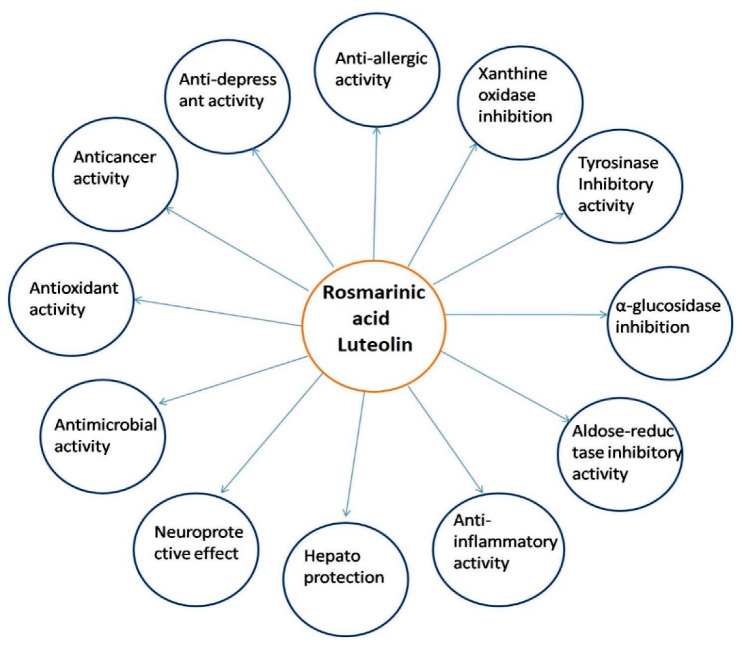
Different biological functions of rosmarinic acid and luteolin, reported from *P. frutescens*.

**Table 1 molecules-27-03578-t001:** Different types of phytoconstituens reported from *P. frutescens*.

S. No	Phytochemical Name	Molecular Formula	Plant Parts	Reference
**Monoterpenes**				
**Acyclic type**				
1.	β-Myrcene	C_10_H_16_	Leaves	[1,2]
2.	Geraniol	C_10_H_18_O	Leaves	[3,4]
3.	β-Citronellene	C_10_H_18_	Leaves	[3]
4.	Nerol	C_10_H_18_O	Leaves	[2]
5.	Linalool	C_10_H_18_O	Leaves, Aerial parts	[4,5]
6.	Ocimene	C_10_H_16_	Leaves	[3]
**Monocyclic type**				
7.	α-Terpineol	C_10_H_18_O	Leaves, Aerial parts	[3,5]
8.	β-Terpineol	C_10_H_18_O	Aerial parts	[6]
9.	Thymol	C_10_H_14_O	Leaves, Aerial parts	[1,5]
10.	α-Phellandrene	C_10_H_16_	Leaves	[1]
11.	β-Phellandrene	C_10_H_16_	Leaves, Aerial parts	[3,6]
12.	1,8-Cineole	C_10_H_18_O	Leaves	[1,2]
13.	Damascenone	C_13_H_18_O	Leaves	[4]
14.	Terpinen-4-ol	C_10_H_18_O	Leaves	[3]
15.	Terpinene	C_10_H_16_	Leaves	[3]
16.	Carvacrol	C_10_H_14_O	Leaves, Aerial parts	[1,5]
17.	Carvone	C_10_H_14_O	Leaves, Aerial parts	[4,7]
18.	Piperitone	C_10_H_16_O	Leaves	[4]
19.	Piperitenone	C_10_H_14_O	Leaves	[3,4]
20.	Limonene	C_10_H_16_	Leaves, Aerial parts	[1,2]
21.	Menthone	C_10_H_18_O	Leaves	[3]
22.	Carveole	C_10_H_16_O	Leaves	[4]
23.	Pulegone	C_10_H_16_O	Leaves	[2]
**Bicyclic type**				
24.	δ -2-Carene	C_10_H_16_	Leaves	[4]
25.	Camphane	C_10_H_18_	Leaves	[3]
26.	Verbenol	C_10_H_16_O	Leaves	[3]
27.	Sabinene	C_10_H_16_	Leaves	[2,3]
**Furanoid type**				
28.	Frutescenone A	C_10_H_10_O_4_	Aerial parts	[8]
29.	Frutescenone B	C_10_H_14_O_3_	Aerial parts	[8]
30.	Frutescenone C	C_15_H_19_N_5_O_2_	Aerial parts	[8]
31.	Isoegomaketone	C_10_H_12_O_2_	Aerial parts	[8,9]
32.	9-hydroxyisoegoma ketone	C_10_H_12_O_3_	Aerial parts	[8,9]
33.	(3S,4R)-3-hydroxy perillaldehyde	C_10_H_14_O	Aerial parts	[8]
34.	Perillic acid	C_10_H_14_O_2_	Aerial parts	[8]
**Sesquiterpenes**				
**Acyclic type**				
35.	α-Farnesene	C_15_H_24_	Leaves, Aerial parts	[2,3,4,5]
36.	β -Farnesene	C_15_H_24_	Leaves, Aerial parts	[3,10]
37.	Farnesol	C_15_H_26_O	Leaves	[3]
38.	Nerolidol	C_15_H_26_O	Leaves, Aerial parts	[2,5]
**Monocyclic type**				
39.	α-Humulene	C_15_H_24_O	Leaves, Aerial parts	[2,5]
40.	Bisabolene	C_15_H_24_	Leaves	[3]
41.	Germacrene	C_15_H_24_	Leaves	[1]
42.	Elemene	C_15_H_24_	Leaves	[1]
43.	β-Ionone	C_13_H_20_O	Leaves, Aerial parts	[1,5]
**Bicyclic type**				
44.	α-pinene	C_10_H_16_	Leaves, Aerial parts	[1,5]
45.	β-pinene	C_10_H_16_	Leaves, Aerial parts	[1]
46.	β-Caryophyllene	C_15_H_24_	Leaves, Aerial parts	[5,11]
47.	ε-Muurolene	C_15_H_24_	Leaves	[3]
48.	α-Cadinene	C_15_H_24_	Leaves	[2]
49.	β-Cadinene	C_15_H_24_	Leaves	[3]
50.	α-Santalol	C_15_H_24_O	Leaves	[3]
51.	α-Bulnesene	C_15_H_24_	Leaves	[3]
52.	β-Gurjunene	C_15_H_24_	Leaves	[3]
53.	β-Selinene	C_15_H_24_	Leaves	[3]
54.	α-Fenchene	C_10_H_16_	Leaves	[3]
55.	α-Cadinol	C_15_H_26_O	Leaves	[12]
56.	Eremophilene	C_15_H_24_	Leaves	[3]
57.	Calarene	C_15_H_24_	Leaves	[3]
58.	Valencene	C_15_H_24_	Aerial parts	[10]
**Tricyclic type**				
59.	Spathulenol	C_15_H_24_O	Leaves, Aerial parts	[1,13]
60.	Viridiflorene	C_15_H_24_	Leaves	[3]
61.	Cubebene	C_15_H_24_	Leaves	[2]
62.	Alloaromadendrene	C_15_H_24_	Leaves	[3]
63.	Patchoulane	C_15_H_26_	Leaves	[3]
64.	α-Copaene	C_15_H_24_	Leaves, Aerial parts	[3,10]
65.	Longifolen	C_15_H_24_	Leaves	[3]
66.	Ylangene	C_15_H_24_	Leaves, Aerial parts	[4]
**Diterpenoids**				
67.	Phytol	C_20_H_40_O	Leaves, Aerial parts	[1,5,11]
**Triterpenoids**				
68.	Ursolic acid	C_30_H_48_O_3_	Leaves	[14,15]
69.	Corosolic acid	C_30_H_48_O_4_	Leaves	[15,16]
70.	3-epicorosolic acid	C_30_H_48_O_4_	Leaves	[15]
71.	Pomolic acid	C_30_H_48_O_4_	Leaves	[15]
72.	Tormentic acid	C_30_H_48_O_5_	Leaves	[15,16]
73.	Hyptadienic acid	C_30_H_46_O_4_	Leaves	[15]
74.	Oleanolic acid	C_30_H_48_O_3_	Leaves	[15]
75.	Augustic acid	C_30_H_48_O_4_	Leaves	[15]
76.	3-epimaslinic acid	C_30_H_48_O_4_	Leaves	[15]
77.	Sericoside	C_36_H_58_O_11_	Leaves	[17]
**Phenyl propanoids**				
78.	Elemicin	C_12_H_16_O_3_	Aerial parts Leaves	[8,18]
79.	Isoelemicin	C_12_H_16_O_3_	Aerial parts	[8]
80.	Myristicin	C_11_H_12_O_3_	Aerial parts Leaves	[8,18]
81.	Eugenol	C_10_H_12_O_2_	Leaves Aerial parts	[1,2]
82.	Isoeugenol	C_10_H_12_O_2_	Leaves	[3,4]
83.	Perilloside E	C_17_H_22_O_9_	Leaves	[19]
84.	Dillapiole	C_12_H_14_O_4_	Leaves	[18]
85.	Nothoapiole	C_13_H_16_O_5_	Leaves	[18]
86.	Allyltetramethoxy benzene		Leaves	[18]
87.	α-Asarone	C_12_H_16_O_3_	Leaves, Aerial Parts	[3,10]
88.	Estragole	C_10_H_12_O	Leaves	[1]
**Alkaloids**				
89.	Neoechinulin A	C_19_H_21_N_3_O_2_	Aerial parts	[8]
90.	1H-indole-3-carboxylic acid	C_9_H_7_NO_2_	Aerial parts	[8]
91.	Indole-3-carboxaldehyde	C_9_H_7_NO	Aerial parts	[8]
**Phenolic compounds**				
92.	Rosmarinic acid	C_18_H_16_O_8_	Leaves, Fruits Seeds	[20,21,22,23]
93.	Methyl rosmarinic acid	C_19_H_18_O_8_	Leaves Seeds	[24,25]
94.	Rosmarinic acid-3-*O*-glucoside	C_24_H_26_O_13_	Seeds	[22,23,24]
95.	3′-dehydroxyl-rosmar inicacid-3-*O*-glucoside	C_24_H_25_O_12_	Seeds	[24]
96.	Caffeic acid	C_9_H_8_O_4_	Leaves, Seeds	[22,24,26]
97.	Ethyl caffeate	C_11_H_12_O_4_	Aerial parts (leaves and stems)	[8]
98.	Methyl caffeate	C_10_H_10_O_4_	Aerial parts (leaves and stems)	[27]
99.	Vinyl caffeate	C_11_H_10_O_4_	Aerial parts (leaves and stems)	[27]
100.	Trans-p-menthenyl caffeate		leaves	[27]
101.	Caffeic acid-3-*O*-glucoside	C_15_H_18_O_9_	Seeds	[23,24]
102.	Protocatechuic acid	C_7_H_6_O_4_	Leaves	[25,28,29]
103.	Protocatechuic aldehyde	C_7_H_6_O_3_	Aerial parts (leaves and stems)	[27]
104.	Chlorogenic acid	C_16_H_18_O_9_	Leaves	[25,29]
105.	Vanillic acid	C_8_H_8_O_4_	Seeds	[24]
106.	Isovanillic acid	C_8_H_8_O_4_	Leaves	[29]
107.	Sinapic acid	C_11_H_12_O_5_	Leaves	[29]
108.	Gallic acid	C_7_H_6_O_5_	Leaves	[29,30]
109.	Ferulic acid	C_10_H_10_O_4_	Leaves Seeds	[29,31]
110.	4-coumaric acid	C_9_H_8_O_3_	Leaves	[32]
111.	Coumaroyl tartaric acid	C_13_H_12_O_8_	Leaves	[30]
112.	4-hydroxyphenyl lactic acid	C_9_H_10_O_4_	Leaves	[33]
113	Sagerinic acid	C_36_H_32_O_16_	Leaves	[16]
114.	Cimidahurinine	C_14_H_20_O_8_	Seeds	[24]
115.	p-Hydroxybenzoic acid	C_7_H_6_O_3_	Leaves	[29]
116.	3,4-DHPEA (Hydroxy tyrosol)	C_8_H_10_O_6_S	Leaves	[30]
**Flavonoids**				
**Flavones**				
117.	Luteolin	C_15_H_10_O_6_	Leaves seeds, fruits	[21,22,23]
118.	Luteolin-7-*O*-glucuronide	C_21_H_18_O_12_	Leaves	[34]
119.	Luteolin-7-*O*-diglucuronide	C_27_H_26_O_18_	Leaves	[32,34]
120.	Luteolin 7-*O*- glucuronide -6”-methyl ester	C_22_H_20_O_12_	Leaves	[35]
121.	Luteolin-5-*O*-glucoside	C_21_H_20_O_11_	Seeds	[24]
122.	Luteolin-7-*O*-glucoside	C_21_H_20_O_11_	Leaves, Seeds	[36,37]
123.	Apigenin	C_15_H_10_O_5_	Leaves, seeds, fruits	[21,22,23,28]
124.	Apigenin-7-*O*-glucuronide	C_21_H_18_O_11_	Leaves	[32]
125.	Apigenin-7-*O*-diglucuronide	C_27_H_26_O_17_	Leaves	[32,34]
126.	Apigenin-7-*O*-glucoside	C_21_H_20_O_10_	Seeds	[37]
127.	Apigenin 7-*O*-caffeoylglucoside	C_30_H_26_O_13_	Leaves	[34]
128.	Diosmetin	C_16_H_12_O_6_	Seeds	[38]
129.	Chrysoeriol	C_16_H_12_O_6_	Fruits, seeds	[21,22]
130.	Scutellarin	C_21_H_18_O_12_	Leaves	[17]
131.	Scutellarein	C_15_H_10_O_6_	Leaves	[17]
132.	Scutellarein -7-*O*-glucuronide	C_21_H_18_O_12_	Leaves	[32,34,39]
133.	Scutellarein 7-*O*-diglucuronide	C_27_H_26_O_18_	Leaves	[32,34,39]
134.	Negletein	C_16_H_12_O_5_	Leaves	[17,28]
135.	Vicenin-2	C_27_H_30_O_15_	Leaves	[40]
136.	Catechin	C_15_H_14_O_6_	Leaves Seeds	[31]
**Flavanones**				
137.	Shisoflavanone A	C_17_H_16_O_5_	Leaves	[28]
138.	Liquiritigenin		Leaves	[16]
139.	5,8-dihydroxy-7-methoxyflavanone	C_16_H_14_O_5_	Leaves	[28]
140.	(2S)-5,7-dimethoxy-8,4′-dihydroxy flavanone	C_17_H_16_O_6_	Leaves	[41]
141.	8-hydroxy-5,7-dimethoxyflavanone	C_17_H_16_O_5_	Leaves	[42]
**Chalcones**				
142.	2′,4′-dimethoxy-4,5′,6′-trihydroxychalcone	C_17_H_16_O_6_	Leaves	[41]
143.	2′,3′-dihydroxy-4′,6′-dimethoxychalcone	C_17_H_16_O_5_	Leaves	[43]
**Aurones**				
144.	(Z)-4,6-dimethoxy-7,4′-dihydroxyaurone	C_17_H_14_O_6_	Leaves	[41]
**Anthocyanins**				
145.	Shisonin (Perillanin) (cyanidin 3-coumaroyl-glucoside- 5-glucoside)	C_36_H_37_O_18_^+^	Leaves	[39,44]
146.	Cis-Shisonin	[Cl^-^]C_36_H_37_O_17_[O^+^]	Leaves	[44]
147.	Cis-malonyl shisonin	C_39_H_39_O_21_^+^	Leaves	[44]
148.	Cyanidin 3-*O*-feruloyl glucoside-5-*O*-glucoside	C_43_H_49_O_24_	Leaves	[44]
149.	Cyanidin 3-*O*-caffeoyl glucoside-5-Oglucoside	C_36_H_37_O_19_^+^	Leaves	[44]
150.	Cyanidin 3-*O*-caffeoyl glucoside-5-*O*-malonyl glucoside	C_30_H_27_O_14_+	Leaves	[44]
151.	Peonidin 3-*O*-malonyl glucoside-5-*O*-p-coumarylglucoside	C_38_H_41_O_17_+	Leaves	[45]
**Coumarins**				
152.	Esculetin	C_9_H_6_O_4_	Leaves	[17,28]
153.	6,7-dihydroxycoumarin	C_9_H_6_O_4_	Leaves and stems	[27]
**Carotenoids**				
154.	Loliolide	C_11_H_16_O_3_	Leaves	[17]
155.	Isololiolide	C_11_H_16_O_3_	Leaves	[17]
**Neolignans**				
156.	Magnosalin	C_24_H_32_O_6_	Leaves	[46]
157.	Andamanicin	C_24_H_36_O_6_	Leaves	[46]
**Glucosides**				
158.	Perillanolide A	C_16_H_26_O_7_	Leaves	[17]
159.	Perillanolide B	C_16_H_26_O_7_	Leaves	[17]
160.	Perilloside A	C_16_H_26_O_6_	Leaves	[19]
161.	Perilloside B	C_16_H_24_O_7_	Leaves	[19]
162.	Perilloside C	C_16_H_28_O_6_	Leaves	[19]
163.	Perilloside E	C_17_H_22_O_9_	Leaves	[19]
164.	Loganin(Iridoid glucoside)	C_17_H_26_O_10_	Leaves	[16]
165.	5’-β-d-glucopyranosyl oxyjasrnonic acid;	C_18_H_28O__9_	Leaves	[16,19]
166.	3-β-d-glucopyrano syl-3-epi-2-isocucur bic acid	C_18_H_30O__8_	Leaves	[19]
167.	3-β-d-glucopyranosyl oxy-5-phenylvaleric acid	C_17_H_24O__8_	Leaves	[19]
168.	N-octanoyl-β-d-fructofuranosyl-α-d-glucopyranoside	C_20_H_36O__12_	Leaves	[16]
169.	Eugenyl-β-d-glucoside	C_16_H_22O__7_	Leaves	[19]
170.	Benzyl-β-d-glucoside	C_13_H_18O__6_	Leaves	[19]
171.	β-sitosteryl β-d-glucoside	C_35_H_60O__6_	Leaves	[19]
172.	Prunasin	C_14_H_17_NO_6_	Leaves	[19]
173.	Sambunigrin	C_14_H_17_NO_6_	Leaves	[19]
**Benzoxepin derivatives**				
174.	Perilloxin	C_16_H_18O__4_	Stems	[47]
175.	Dehydroperilloxin	C_16_H_16O__4_	Stems	[47]
**Policosanols**				
176.	Eicosnaol	C_20_H_42O_	Seeds	[48,49,50]
177.	Heneicosanol	C_21_H_44O_	Seeds	
178.	Docosanol	C_22_H_46O_	Seeds	
179.	Tricosanol	C_23_H_48O_	Seeds	
180.	Tetracosanol	C_24_H_50O_	Seeds	
181.	Pentacosanol	C_25_H_52O_	Seeds	
182.	Hexacosanol	C_26_H_54O_	Seeds	
183.	Heptacosanol	C_27_H_56O_	Seeds	
184.	Octacosanol	C_28_H_58O_	Seeds	
185.	Nonacosanol	C_29_H_60O_	Seeds	
186.	Triacontanol	C_30_H_62O_	Seeds	
**Phytosterols**				
187.	Stigmasterol	C_29_H_48O_	Seeds	[50,51]
188.	β-sitosterol	C_31_H_52O__2_	Seeds	[50,51]
189.	Campesterol	C_28_H_48O_	Seeds	[50]
190.	β-amyrin	C_30_H_50O_	Seeds	[50]
191.	β-cholestanol	C_27_H_48O_	Seeds	[50]
192.	5α-cholestane	C_27_H_48_	Seeds	[50,52]
**Tocopherols**				
193.	δ-tocopherol	C_27_H_46O__2_	Seeds	[50,51]
194.	γ-tocopherol	C_28_H_48O__2_	Seeds	
195.	β-tocopherol	C_28_H_48O__2_	Seeds	
196.	α-tocopherol	C_29_H_50O__2_	Seeds	
**Fatty acids**				
197.	Lauric acid	C_12_H_24_O2	Seeds	[53]
198.	Myristic acid	C_14_H_28_O2	Seeds	[53]
199.	Pentadecanoic acid	C_15_H_30_O2	Seeds	[50]
200.	Palmitic acid	C_16_H_32_O2	Seeds	[50]
201.	Palmitoleic acid	C_16_H_30_O2	Seeds	[53]
202.	Heptadecanoic acid	C_17_H_34_O2	Seeds	[53]
203.	Stearic acid	C_18_H_36_O2	Seeds	[50]
204.	Oleic acid	C_18_H_34_O2	Seeds	[50]
205.	Linoleic acid	C_18_H_32_O2	Seeds	[50]
206.	Linolenic acid	C_18_H_30_O2	Seeds	[50]
207.	Arachidic acid	C_20_H_40_O_2_	Seeds	[50]
208.	Eicosenoic acid	C_20_H_38_O_2_	Seeds	[50]
209.	Eicosadienoic acid	C_20_H_36_O_2_	Seeds	[53]
210.	Eicosatrienoic acid	C_20_H_34_O_2_	Seeds	[53]
211.	Behenic acid	C_22_H_44_O2	Seeds	[53]
**Other important compounds**				
212.	p-Hydroxybenzaldehyde	C_7_H_6_O_2_	Leaves	[17]
213.	p-Hydroxyacetophenone	C_8_H_8_O_2_	Leaves	[17]
214.	trans-p-Hydroxycinnamic acid	C_9_H_8_O_3_	Leaves	[17]
215.	3′,4′,5′-trimethoxycinnamyl alcohol	C_12_H_16_O_4_	Aerial parts	[8]

**Table 2 molecules-27-03578-t002:** Different volatile components present in the leaves and aerial parts of *P. frutescens*.

S. No.	Component	Mol. Formula	Parts	Reference
1.	α-Farnesene	C_15_H_24_	Leaves, Aerial parts	[3,5,11]
2.	β -Farnesene	C_15_H_24_	Leaves, Aerial parts	[3,10]
3.	α-Caryophyllene	C_15_H_24_	Leaves, Aerial parts	[3,7]
4.	β-Caryophyllene	C_15_H_24_	Leaves, Aerial parts	[5,11]
5.	Isocaryophyllene	C_15_H_24_	Leaves, Aerial parts	[3,10]
6.	Caryophyllene oxide	C_15_H_24_O	Leaves, Aerial parts	[1,3,5]
7.	Phytol	C_20_H_40_O	Leaves, Aerial parts	[1,5,11]
8.	Alloaromadendrene	C_15_H_24_	Leaves	[3]
9.	Thymoquinone	C_10_H_12_O_2_	Leaves	[11,57]
10.	Bergamotene	C_16_H_24_	Leaves, Aerial parts	[4,11]
11.	Diisooctyl adipate	C_22_H_42_O_4_	Leaves	[11]
12.	α-pinene	C_10_H_16_	Leaves, Aerial parts	[1,5]
13.	β-pinene	C_10_H_16_	Leaves, Aerial parts	[1,5]
14.	Eugenol	C_10_H_12_O_2_	Leaves, Aerial parts	[1,2,7]
15.	Isoeugenol	C_10_H_12_O_2_	Leaves	[3,4]
16.	Methyl eugenol	C_11_H_14_O_2_	Leaves	[3]
17.	Methyl isoeugenol	C_11_H_14_O_2_	Leaves, Aerial parts	[3,8]
18.	Spathulenol	C_15_H_24_O	Leaves, Aerial parts	[1,5,13]
19.	Viridiflorene	C_15_H_24_	Leaves	[3]
20.	Viridiflorol	C_15_H_26_O	Leaves	[3]
21.	ε-Muurolene	C_15_H_24_	Leaves	[3]
22.	γ-Terpinene	C_10_H_16_	Leaves	[2]
23.	β-Terpinene	C_10_H_16_	Leaves	[3]
24.	Terpinen-4-ol	C_10_H_18_O	Leaves	[3]
25.	Nerolidol	C_15_H_26_O	Leaves, Aerial parts	[2,5]
26.	α-Cadinene	C_15_H_24_	Leaves	[2]
27.	β-Cadinene	C_15_H_24_	Leaves	[3]
28.	δ-Cadinene	C_15_H_24_	Leaves, Aerial parts	[3,5]
29.	α-Asarone	C_12_H_16_O_3_	Leaves, Aerial parts	[3,10]
30.	Linalool	C_10_H_18_O	Leaves, Aerial parts	[4,5]
31.	Linalool propanoate	C_14_H_24_O_2_	Leaves	[1]
32.	Linalool formate	C_11_H_18_O_2_	Leaves	[1]
33.	Linalool oxide	C_10_H_18_O_2_	Leaves	[58]
34.	Carvacrol	C_10_H_14_O	Leaves, Aerial parts	[1,5]
35.	α-Santalol	C_15_H_24_O	Leaves	[3]
36.	α-Bulnesene	C_15_H_24_	Leaves	[3]
37.	β-Gurjunene	C_15_H_24_	Leaves	[3]
38.	β-Selinene	C_15_H_24_	Leaves	[3]
39.	Germacrene A	C_15_H_24_	Leaves	[1]
40.	Germacrene B	C_15_H_24_	Leaves	[1]
41.	Germacrene D	C_15_H_24_	Leaves, Aerial parts	[7,12]
42.	Bicyclogermacrene	C_15_H_24_	Leaves	[2]
43.	Estragole	C_10_H_12_O	Leaves	[1]
44.	α-Cubebene	C_15_H_24_	Leaves	[2]
45.	β-Cubebene	C_15_H_24_	Leaves	[2]
46.	Carvone	C_10_H_14_O	Leaves, Aerial parts	[4,7]
47.	Shisofuran	C_10_H_12_O	Leaves	[1,4]
48.	Piperitone	C_10_H_16_O	Leaves	[4]
49.	Piperitenone	C_10_H_14_O	Leaves	[3,4]
50.	Farnesol	C_15_H_26_O	Leaves	[3]
51.	Phytone	C_18_H_36_O	Leaves	[3]
52.	α-citral	C_10_H_16_O	Leaves	[3]
53.	β-citral	C_10_H_16_O	Leaves	[1]
54.	Ocimene	C_10_H_16_	Leaves	[3]
55.	Cosmene	C_10_H_14_	Leaves	[3]
56.	γ-Pyronene	C_10_H_16_	Leaves	[3]
57.	Perillene	C_10_H_14_O	Leaves, Aerial parts	[3,13]
58.	Perillaldehyde	C_10_H_14_O	Leaves, Aerial parts	[1,3,5]
59.	Perilla ketone	C_10_H_14_O_2_	Leaves, Aerial parts	[1,13]
60.	Egoma ketone	C_10_H_12_O_2_	Leaves, Aerial parts	[5,12]
61.	Isoegomaketone	C_10_H_12_O_2_	Leaves, Aerial parts	[12,13]
62.	Perilla alcohol	C_10_H_16_O	Leaves, Aerial parts	[1,2,5]
63.	Perillic acid	C_10_H_14_O_2_	Aerial parts	[5]
64.	Methy perillate	C_11_H_16_O_2_	Aerial parts	[5]
65.	Elscholtzia ketone	C_10_H_14_O_2_	Leaves, Aerial parts	[1,13]
66.	dehydro-elsholtzia ketone	C_10_H_12_O_2_	Leaves, Aerial parts	[1]
67.	Naginata ketone		Leaves	[3]
68.	α-Terpineol	C_10_H_18_O	Leaves, Aerial parts	[3,5]
69.	Elemicin	C_12_H_16_O_3_	Leaves, Aerial parts	[3,10]
70.	Isoelemicin	C_12_H_16_O_3_	Leaves, Aerial parts	[3,8]
71.	Myristicin	C_11_H_12_O_3_	Leaves, Aerial parts	[10]
72.	Dillapiol	C_12_H_14_O_4_	Leaves	[18]
73.	Nothoapiol	C_13_H_16_O_5_	Leaves	[18]
74.	Patchoulane	C_15_H_26_	Leaves	[3]
75.	α-Patchoulene	C_15_H_24_	Leaves	[3]
76.	o-Cymene	C_10_H_14_	Aerial parts	[7]
77.	p-Cymene	C_10_H_14_	Aerial parts	[5]
78.	Pulegone	C_10_H_16_O	Leaves	[2]
79.	Isopulegone	C_10_H_16_O	Leaves	[3]
80.	β-Bourbonene	C_15_H_24_	Leaves	[3]
81.	α-Humulene	C_15_H_24_	Leaves, Aerial parts	[2,5]
82.	Humulene epoxide II	C_15_H_24_O	Leaves, Aerial parts	[3,5]
83.	α-Bisabolene epoxide	C_15_H_24_O	Leaves	[3]
84.	Isoaromadendrene epoxide	C_15_H_24_O	Aerial parts	[6]
85.	Sabinene	C_10_H_16_	Leaves	[2,3]
86.	Styrene	C_8_H_8_	Leaves	[3]
87.	Limonene	C_10_H_16_	Leaves, Aerial parts	[2,10]
88.	Limonene oxide	C_10_H_16_O	Leaves	[2]
89.	Limonene aldehyde	C_11_H_18_O	Leaves	[1]
90.	Isolimonene	C_10_H_16_O	Leaves	[3]
91.	Pseudolimonene	C_10_H_16_	Leaves, Aerial parts	[3,10]
92.	α-Copaene	C_16_H_26_	Leaves, Aerial parts	[3,10]
93.	α-Fenchene	C_10_H_16_	Leaves	[3]
94.	Anisole	C_7_H_8_O	stems	[59]
95.	Eucalyptol	C_10_H_18_O	Leaves, Aerial parts	[3,6]
96.	β-Myrcene	C_10_H_16_	Leaves	[1,2]
97.	Geranyl acetone	C_13_H_22_O	Leaves	[1,2]
98.	Hexahydrofarnesyl acetone	C_18_H_36_O	Leaves, Aerial parts	[5,13]
99.	Methyl geranate	C_11_H_18_O_2_	Leaves	[1,2]
100.	Camphene	C_10_H_16_	Leaves	[1,2]
101.	Longifolen	C_15_H_24_	Leaves	[3]
102.	1,8-Cineole	C_10_H_18_O	Leaves	[1,2]
103.	Damascenone	C_13_H_18_O	Leaves	[4]
104.	α-Cadinol	C_15_H_26_O	Leaves	[12]
105.	Tau-Cadinol	C_15_H_26_O	Leaves	[1]
106.	α-Terpinolene	C_10_H_16_	Leaves	[1,2]
107	Menthol	C_10_H_20_O	Leaves	[3]
108.	Menthone	C_10_H_18_O	Leaves	[3]
109.	Isomenthone	C_10_H_18_O	Leaves	[3]
110.	Eremophilene	C_15_H_24_	Leaves	[3]
111.	Carveole	C_10_H_16_O	Leaves	[4]
112.	Dihydrocarveol	C_10_H_18_O	Leaves, Aerial parts	[3,5]
113.	Dihydrocarveol acetate	C_12_H_20_O_2_	Leaves	[1,3]
114.	Isodihydrocarveol acetate	C_12_H_20_O_2_	Leaves	[58]
115.	Geraniol	C_10_H_18_O	Leaves	[3,4]
116.	α-Terpineol	C_10_H_18_O	Leaves, Aerial parts	[3,5]
117.	β-Terpineol	C_10_H_18_O	Aerial parts	[6]
118.	β-Elemene	C_15_H_24_	Leaves	[1,2]
119.	δ -Elemene	C_15_H_24_	Leaves	[2,3]
120.	β-Citronellene	C_10_H_18_	Leaves	[3]
121.	δ -2-Carene	C_10_H_16_	Leaves	[4]
122.	Calarene	C_15_H_24_	Leaves	[3]
123.	Camphane	C_10_H_18_	Leaves	[3]
124.	Ylangene	C_15_H_24_	Leaves, Aerial parts	[4]
125.	Nerol	C_10_H_18_O	Leaves	[2]
126.	Cadina 3,9-diene	C_15_H_24_	Aerial parts	[7]
127.	Neophytadiene	C_20_H_38_	Leaves	[12]
128.	β-Ionone	C_13_H_20_O	Leaves, Aerial parts	[1,5]
129.	α -Fenchene	C_10_H_16_	Leaves	[3]
130.	Thymol	C_10_H_14_O	Leaves, Aerial parts	[1,5]
131.	α-Phellandrene	C_10_H_16_	Leaves	[1]
132.	β-Phellandrene	C_10_H_16_	Leaves, Aerial parts	[3,6]
133.	Santolina triene	C_10_H_16_	Leaves	[3]
134.	Verbenol	C_10_H_16_O	Leaves	[3]
135.	trans-Shisool	C_10_H_18_O	Leaves, Aerial parts	[1,6]
136.	Thujyl alcohol	C_10_H_18_O	Leaves	[3]
137.	Furfuryl alcohol	C_5_H_6_O_2_	Leaves	[3]
138.	2-Hexanoylfuran	C_10_H_14_O_2_	Leaves, Aerial parts	[10,11]
139.	2-acetylfuran	C_6_H_6_O_2_	Leaves	[3]
140.	β-terpinyl acetate	C_12_H_20_O_2_	Aerial parts	[7]
141.	trans-Valerenyl acetate	C_17_H_26_O_2_	Leaves	[11]
142.	Isomenthyl acetate	C_12_H_22_O_2_	Leaves	[1]
143.	Isobornyl acetate	C_12_H_20_O_2_	Aerial parts	[6]
144.	Bornyl acetate	C_12_H_20_O_2_	Leaves	[58]
145.	Nerol acetate	C_12_H_20_O_2_	Leaves	[3]
146.	2-furyl methyl ketone	C_6_H_6_O_2_	Aerial parts	[7]
147.	Valencene	C_15_H_24_	Aerial parts	[10]
148.	laurolene	C_8_H_14_	Leaves	[3]
149.	α-curcumene	C_15_H_22_	Stems	[59]
150.	Elixene	C_15_H_24_	Leaves	[59]
151.	Curlone	C_15_H_22_O	Stems	[59]
152.	Isopiperitenol	C_10_H_16_O	Leaves	[60]
153.	Isopiperitenone	C_10_H_14_O	Leaves	[60]
154.	Neral	C_10_H_16_O	Leaves	[60]
155.	Geranial	C_10_H_16_O	Leaves	[60]
156.	Geraniol	C_10_H_18_O	Leaves	[60]
157.	α-naginatene	C_10_H_14_O	Leaves	[60]
158.	β -cyclocitral	C_10_H_16_O	Leaves	[58]
159.	Pthalic acid	C_8_H_6_O_4_	Stems	[59]
160.	2-Butylamine	C_4_H_11_N	Leaves	[3]
161.	2-Pyrimidinamine	C_4_H_5_N_3_	Aerial parts	[10]
162.	2-Hydroxypyridine	C_5_H_5_NO	Leaves	[3]
163.	Phenylacetaldehyde	C_8_H_8_O	Leaves	[12]
164.	p-Mentha-3,8-diene	C_10_H_16_	Leaves	[3]
165.	Methyl thymyl ether	C_11_H_16_O	Seeds	[59]
166.	p-mentha-2,4(8)-diene	C_10_H_16_	Leaves, Aerial parts	[4,7]
167.	p-Menth-2-en-1-ol	C_10_H_18_O	Leaves	[2]
168.	p-Menth-1-en-8-ol	C_10_H_18_O	Leaves, Aerial parts	[2,4,7]
169.	p-Mentha-1,8-dien-7-ol	C_10_H_16_O	Leaves, Aerial parts	[2,7]
170.	p-Menth-4(8)-en-3-one	C_10_H_16_O	Leaves	[2]
171.	2-Butanone	C_4_H_8_O	Leaves	[2]
172.	1-Pentene-3-one	C_5_H_8_O	Leaves	[12]
173.	3-Pentanone	C_5_H_10_O	Leaves	[12]
174.	2-Cyclopentenone	C_5_H_6_O	Leaves	[3]
175.	4,4-Dimethyl-2-cyclopenten-1-one	C_7_H_10_O	Leaves	[3]
176.	2-Methylcyclopentanone	C_6_H_10_O	Leaves	[3]
177.	2-Methyl-2-cyclopentenone		Leaves	[3]
178.	Cyclohexanone	C_6_H_8_O	Leaves	[3]
179.	Methyl heptenone	C_8_H_14_O	Leaves	[3]
180.	1-octen-3-one	C_8_H_14_O	Leaves	[12]
181.	1-Pentene-3-ol	C_5_H_10_O	Leaves	[12]
182.	2-Pentenol	C_5_H_10_O	Leaves	[12]
183.	2-Hexenol	C_6_H_12_O	Leaves	[12]
184.	3-Hexenol	C_6_H_12_O	Leaves	[12]
185.	1-Hexanol	C_6_H_14_O	Leaves	[12]
186.	1-Octen-3-ol	C_8_H_16_O	Leaves, Aerial parts	[3,4,13]
187.	3-Octanol	C_8_H_18_O	Leaves, Aerial parts	[4,10]
188.	Octadienol	C_8_H_14_O	Leaves	[2]
189.	Benzaldehyde	C_7_H_6_O	Leaves, Aerial parts	[2,5,13]
190.	3-Pentenal	C_5_H_8_O	Leaves	[2,12]
191.	Hexanal	C_6_H_12_O	Leaves	[2,12]
192.	2-Hexenal	C_6_H_10_O	Leaves	[2,12]
193.	3-Hexenal	C_6_H_10_O	Leaves	[2,12]
194.	2,4-Hexadienal	C_6_H_8_O	Leaves	[2,12]
195.	2,4-Heptadienal	C_7_H_10_O	Leaves	[2,12]
196.	Octanal	C_8_H_16_O	Leaves	[2,12]

## Data Availability

Not applicable.

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
