# Peer review of "Perilla frutescens: A Rich Source of Pharmacological Active Compounds"

_molecules, 2022, doi:10.3390/molecules27113578_

Round 1

Reviewer 1 Report

This is a well-written review article with extensive details on the Perilla frutescens. In addition, this article explains well the medicinal value of the Perilla frutescens plant.

However, I have a few questions,
Is there any food toxicity with this plant, or are there any side effects?
comments
Paragraph 3.3 has two different font sizes or font styles; please correct it.

Author Response

Reviewer 1:

Comments and Suggestions for Authors

This is a well-written review article with extensive details on the Perilla frutescens. In addition, this article explains well the medicinal value of the Perilla frutescens plant.

However, I have a few questions,

Is there any food toxicity with this plant, or are there any side effects?

Perilla frutescens shows no toxicity and no side effects to human. However, it shows little side effects to cattle and livestock. But this side effect is negligible when the cattle eat limited P. frutescens. Perilla frutescens contains a pneumotoxin in the leaves and seeds that when metabolized in the rumen produces toxic intermediaries. Perilla ketone, from the essential oil of Perilla frutescens, is a pulmonary edemagenic agent for laboratory animals and livestock when they take high amount of P. frutescens. (Page 35, Last Paragraph, Text Highlighted with Yellow Color).

comments

Paragraph 3.3 has two different font sizes or font styles; please correct it.

We have corrected this paragraph. The paragraph has only one font size and style. Palatino Linotype , 10 (Paragraph 3.3 , Page 33-34)

Reviewer 2 Report

1- Perilla frutescens as a species sure it possesses synonyms did the authors review all  synonyms of plant taxa?:

For example for the Plant list database (http://www.theplantlist.org/tpl/search?q=Perilla+frutescens), the species is known under 4 Accepted scientific names  

3- According to kewscience (https://wcsp.science.kew.org/synonomy.do;jsessionid=7012191E89CE008C351132487A43C8AE.kppapp05-wcsp?name_id=150299) the plant species is known under 4 different scientific names ( Ocimum frutescens L., Sp. Pl.: 597 (1753).

3- phytochemicals of the species including alkaloids, terpenoids, quinines, phenylpropanoids, polyphenolic compounds,.......ect have
been reported. I suggest the authors summarize chemical structures with the background in depth  about the metabolism of chemical families.

4- including alkaloids, terpenoids, quinines, phenylpropanoids,  polyphenolic (NOT polyphenoilc) compounds, 

Author Response

Reviewer 2:

Comments and Suggestions for Authors

1- Perilla frutescens as a species sure it possesses synonyms did the authors review all synonyms of plant taxa?:

For example for the Plant list database (http://www.theplantlist.org/tpl/search?q=Perilla+frutescens), the species is known under 4 Accepted scientific names  

Ans : According to the plant list database, Perilla frutescens has 13 synonyms. Among these Perilla frutescens (L.) Britton is widely accepted with the highest confidence level (***). Further in all the research publications and research databases, all the researchers or scientists use the name Perilla frutescens (L.) Britton.  Hence we include Perilla frutescens (L.) Britton in our review.

For reference, we include the plant list data base image below.

 According to kewscience

(https://wcsp.science.kew.org/synonomy.do;jsessionid=7012191E89CE008C351132487A43C8AE.kppapp05-wcsp?name_id=150299) the plant species is known under 4 different scientific names ( Ocimum frutescens L., Sp. Pl.: 597 (1753).

Ans : According to the kewscience database, Perilla frutescens has 4 synonyms/homotypic names including

Ocimum frutescens L., Sp. Pl.: 597 (1753).

Perilla ocymoides L., Gen. Pl. ed. 6: 578 (1764), nom. superfl.

Perilla urticifolia Salisb., Prodr. Stirp. Chap. Allerton: 80 (1796), nom. superfl.

Perilla frutescens var. typica Makino, J. Jap. Bot. 3: 7 (1926), not validly publ.

But the accepted name is Perilla frutescens (L.) Britton. Hence we use this accepted name throughout the manuscript.

3- phytochemicals of the species including alkaloids, terpenoids, quinines, phenylpropanoids, polyphenolic compounds,.......ect have been reported. I suggest the authors summarize chemical structures with the background in depth  about the metabolism of chemical families.

Ans : The present review discusses the phytochemicals present in P. frutescens and their biological functions. In the present review we have mentioned about 400 different phytochemicals present in P. frutescens. Further we mentioned the maximum plausible biological functions of P. frutescens with in deapth  mechanisms.  Further, the present review is really large one as it contains almost 42 pages of manuscript with about 18000 words. Hence it is difficult to include structures as well as the metabolic fate of chemical families. Further all the chemical structures are easily available to researchers in pubchem database. Hence we think, it is not necessary to include structures in the review.

4- including alkaloids, terpenoids, quinines, phenylpropanoids,  polyphenolic (NOT polyphenoilc) compounds, 

Ans : The Spelling mistake is corrected (Refer Page 1, Abstract Section, Line Number 8, Highlighted with Yellow)

Reviewer 2:

Comments and Suggestions for Authors

1- Perilla frutescens as a species sure it possesses synonyms did the authors review all synonyms of plant taxa?:

For example for the Plant list database (http://www.theplantlist.org/tpl/search?q=Perilla+frutescens), the species is known under 4 Accepted scientific names  

Ans : According to the plant list database, Perilla frutescens has 13 synonyms. Among these Perilla frutescens (L.) Britton is widely accepted with the highest confidence level (***). Further in all the research publications and research databases, all the researchers or scientists use the name Perilla frutescens (L.) Britton.  Hence we include Perilla frutescens (L.) Britton in our review.

For reference, we include the plant list data base image below.

 According to kewscience

(https://wcsp.science.kew.org/synonomy.do;jsessionid=7012191E89CE008C351132487A43C8AE.kppapp05-wcsp?name_id=150299) the plant species is known under 4 different scientific names ( Ocimum frutescens L., Sp. Pl.: 597 (1753).

Ans : According to the kewscience database, Perilla frutescens has 4 synonyms/homotypic names including

Ocimum frutescens L., Sp. Pl.: 597 (1753).

Perilla ocymoides L., Gen. Pl. ed. 6: 578 (1764), nom. superfl.

Perilla urticifolia Salisb., Prodr. Stirp. Chap. Allerton: 80 (1796), nom. superfl.

Perilla frutescens var. typica Makino, J. Jap. Bot. 3: 7 (1926), not validly publ.

But the accepted name is Perilla frutescens (L.) Britton. Hence we use this accepted name throughout the manuscript.

3- phytochemicals of the species including alkaloids, terpenoids, quinines, phenylpropanoids, polyphenolic compounds,.......ect have been reported. I suggest the authors summarize chemical structures with the background in depth  about the metabolism of chemical families.

Ans : The present review discusses the phytochemicals present in P. frutescens and their biological functions. In the present review we have mentioned about 400 different phytochemicals present in P. frutescens. Further we mentioned the maximum plausible biological functions of P. frutescens with in deapth  mechanisms.  Further, the present review is really large one as it contains almost 42 pages of manuscript with about 18000 words. Hence it is difficult to include structures as well as the metabolic fate of chemical families. Further all the chemical structures are easily available to researchers in pubchem database. Hence we think, it is not necessary to include structures in the review.

4- including alkaloids, terpenoids, quinines, phenylpropanoids,  polyphenolic (NOT polyphenoilc) compounds, 

Ans : The Spelling mistake is corrected (Refer Page 1, Abstract Section, Line Number 8, Highlighted with Yellow)

Reviewer 3 Report

The manuscript gives a useful overview of the chemical constituents and pharmaceutical activities of Perilla frutescens, an important pharmaceutical and nutraceutical crop widely cultivated in Asian countries and other regions of the globe.

The rationale for the establishment of this review was nicely founded. This ascertains its novelty and its importance in the context of the review. Furthermore, the summary of the researches seems very rich in the content.
I have read the review thoroughly and did not find any substantive allegation in the text.

I would suggest it for publication, but I could not find the cited Figures and Tables, so it is impossible to fulfill the review at this stage. I enclose the only file that I was able to download for proof of this unusual situation. The Authors should enclose this important part of the manuscript.

Furthermore, there are some minor but numerous linguistic errors and misspellings that need to be addressed by the professional corrector.

Author Response

Reviewer 3:

Comments and Suggestions for Authors

The manuscript gives a useful overview of the chemical constituents and pharmaceutical activities of Perilla frutescens, an important pharmaceutical and nutraceutical crop widely cultivated in Asian countries and other regions of the globe.

The rationale for the establishment of this review was nicely founded. This ascertains its novelty and its importance in the context of the review. Furthermore, the summary of the researches seems very rich in the content.

I have read the review thoroughly and did not find any substantive allegation in the text.

I would suggest it for publication, but I could not find the cited Figures and Tables, so it is impossible to fulfill the review at this stage. I enclose the only file that I was able to download for proof of this unusual situation. The Authors should enclose this important part of the manuscript.Furthermore, there are some minor but numerous linguistic errors and misspellings that need to be addressed by the professional corrector.

Ans : The figures and Tables were included in the text in the revised manuscript. The language was checked and corrected by professional corrector. The English Editing Certificate was attached.

Round 2

Reviewer 3 Report

Authors corrected the manusript according to my suggestions. The tables and figures that were missing were inserted into the manuscript's text. They are relevant and well prepared. Manusript was improved linguistically.  In this form, the work can be suggested for publication.

This manuscript is a resubmission of an earlier submission. The following is a list of the peer review reports and author responses from that submission.